# LARGE LEGISLATIVE MODELS:
# TOWARDS EFFICIENT AI POLICYMAKING IN ECONOMIC SIMULATIONS

## ABSTRACT

The improvement of economic policymaking presents an opportunity for broad societal benefit, a notion that has inspired research towards AI-driven policymaking tools. AI policymaking holds the potential to surpass human performance through the ability to process data quickly at scale. However, existing RL-based methods exhibit sample inefficiency, and are further limited by an inability to flexibly incorporate nuanced information into their decision-making processes. Thus, we propose a novel method in which we instead utilize pre-trained Large Language Models (LLMs), as sample-efficient policymakers in socially complex multi-agent reinforcement learning (MARL) scenarios. We demonstrate significant efficiency gains, outperforming existing methods across three environments.

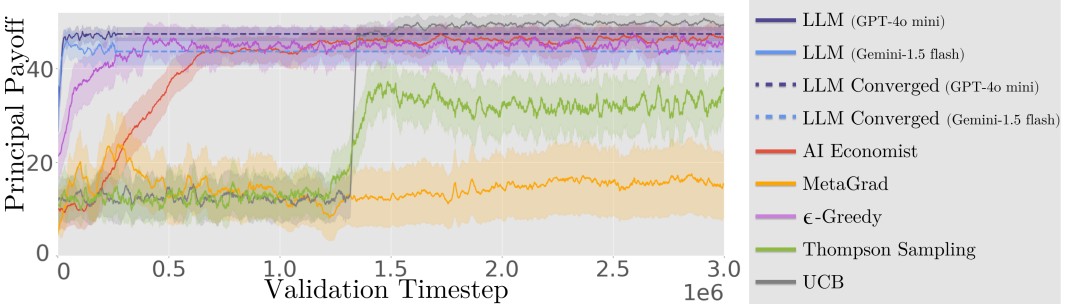

Figure 1: Comparison of policymaker (principal) performance in the Commons Harvest Open environment. Our method demonstrates superior sample efficiency over all existing methods. Each method is run on 10 seeds.

## 1 INTRODUCTION

Economic policy-making is a field rife with uncertainty (Al-Thaqeb & Algharabali, 2019), high stakes (Persson & Tabellini, 2004), and complexity (Mueller, 2020). Human policy-makers are often faced with overwhelming amounts of data (Huh et al., 2018) and the influence of vested interests (Elliott, 1997), complicating effective and equitable decision-making. AI-driven tooling, with the ability to avoid self-centered bias and parse large amounts of data quickly, could offer significant assistance.

Existing research (Zheng et al., 2022; Yang et al., 2022) towards AI-based policymaking has primarily focused on reinforcement learning (RL) based methods using neural networks as economic policy generators [1]. A leading method in this field, termed AI Economist (Zheng et al., 2022), attempts to maximize the social welfare of agents within a MARL environment called Gather-Trade-Build (GTB). In GTB, agents can move around, collect wood and stone, and trade resources in order to gain reward. AI Economist employs a neural network that observes agent endowments and market

---

[1]Note that the word "policy" is overloaded. We use "economic policy" to refer to decisions made by a social planner, such as tax rates, synonymous with "fiscal policy". In contrast, we use "policy" to refer to the economic agents' strategy, synonymous with "reinforcement learning policy".

behavior, and outputs tax rates; for a fixed set of weights, this network is an economic policy generator – a function mapping observed economic data to economic policy, such as tax rates. Policymaking unfolds as a bilevel optimization problem, where the outer loop trains the network's weights, and each set of weights induces an economic policy generator parameterizing the environment that agents train against in the inner loop.

Though environment observations like agent endowments may contain useful information for the outer loop generator optimization, they are not necessarily variables that economic policies should be a function of. Notably, we found that removing the generator network observations from both the AI Economist and Yang et al. (2022)'s MetaGrad, another leading method, *had no effect on their performance*. This implies that optimal tax rates can be static in their environments, and agent endowments or market behavior are not needed as input. However, such observations should inform the outer policymaking process. For AI Economist and MetaGrad, this information is limited to being used through the generator network optimization step, where observations from an episode affect the subsequent update to generator network weights only through their gradient in the loss. Our ablations demonstrate this is an overly complicated way to approach AI policymaking – a problem that exacerbates the sample inefficiency that RL methods already suffer from. In our test environments, both methods suffered from a starkly visible sample inefficiency problem; AI Economist was outperformed or matched by a much simpler $\epsilon$-greedy bandit algorithm on two of our three environments, and MetaGrad by all three bandit baselines we use across all test environments.

Motivated by these limitations, we propose a simpler alternative approach to AI-based policymaking by leveraging pre-trained Large Language Models (LLMs) as policymakers. Instead of learning economic policy generators, we directly learn economic policy by applying the In-Context Learning (ICL) (Dong et al., 2024) capabilities of LLMs. In addition, as our method uses a sequence modeling approach to predict policies as prompt completions, it is highly flexible in its inputs. For example, an economic report from human experts detailing suggested tax rates could improve sample efficiency in finding an optimal tax rate policy. For an LLM, the entire report in natural language can be added to a prompt to inform the policy subsequently chosen, while it is unclear how this would be done for existing methods without further augmentation. Furthermore, LLMs can also be given a contextualization of the problem setting they are applied to, potentially allowing them to draw upon their extensive pre-training data distribution to improve sample efficiency and solve more complex, realistic environments. Overall, our method is significantly more sample efficient than prior approaches. We outperform five baselines significantly in terms of sample efficiency across three multi-agent test environments, with little compromise to final asymptotic performance.

In summary, our contributions are as follows: (1) We give a generalized formalization of AI-based policymaking methods, and leverage it to analyze the limitations of existing methods. (2) We propose a novel LLM-based automated policymaking method that addresses sample inefficiency in prior approaches and simplifies the policymaking process. (3) We provide extensive empirical results to demonstrate the efficacy of our approach and analyze the contribution of each component to its overall performance.

## 2 PRELIMINARIES

We follow Zhang et al. (2024) in using Stackelberg-Markov games as the foundational concept from which to construct formalizations of economic policy-making.

**Definition 2.1.** A **Partially Observable Markov Game** (POMG) $\mathcal{M}$ with $n$ agents is a tuple $(S, A, T, r, \Omega, O, \gamma, \mu_0)$, where $S$ is a state space, $A = \times_{i \in [n]} A_i$ is the joint action space, $T : S \times A \mapsto \Delta(S)$ is a stochastic transition function, $r : S \times A \mapsto \mathcal{R}^n$ is the reward function $r = (r_i)_{i=1}^{n}$, $\Omega = \times_{i \in [n]} \Omega_i$ is the joint observation space, $O : S \times A \mapsto \Delta(\Omega)$ is a stochastic observation function, $\gamma \in [0, 1)$ is a discount factor, and $\mu_0 \in \Delta(S)$ is the initial state distribution. In episodes of this game, agent behavior is characterized by policies $\pi_i : \Omega_i \mapsto A$.

**Definition 2.2.** An $(n + 1)$-player **Stackelberg-Markov Game** $\mathcal{S} = (n, \Phi, \Pi, \mathbf{u})$, consists of a principal (policymaker) and $n$ followers. At each episode, the principal commits to an action $\phi \in \Phi$ from action space $\Phi$, which induces an $n$-player POMG $\mathcal{M}^\phi = \left(S, A^\phi, T^\phi, r^\phi, \Omega^\phi, O^\phi, \gamma, \mu_0^\phi\right)$. Having observed $\phi$, each follower $i \in [n]$ responds with a policy $\pi_i : \Omega_i \mapsto A$ in their policy space $\Pi_i$. The follower joint policy space is $\Pi = \times_{i \in [n]} \Pi_i$. After all players choose an action,

the leader receives payoff $u_0(\phi, \pi) \in \mathbb{R}$, while each follower $i \in [n]$ receives payoff $u_i(\phi, \pi) = \mathbb{E}^{\mathcal{M}^\phi, \pi}\left[\sum_{t=0}^{\infty} \gamma^t r_i^\phi(s^t, a^t)\right] \in \mathbb{R}$. Note $u_0(\phi, \pi)$ is usually set to $\sum_i u_i(\phi, \pi)$.

Zhang et al. (2024) define equilibria in Stackelberg-Markov games using a nested maximization condition: an action $\phi^*$ maximizing the principal's payoff, evaluated on a follower joint policy $\pi^*$ drawn from a best-response set. We instead extend the work of Gerstgrasser & Parkes (2023) on oracle abstractions in Markov games to the setting of Stackelberg-Markov games.

**Definition 2.3.** We denote as $\mathcal{E}(\mathcal{M}^\phi)$ a distribution over follower equilibria in a Partially Observable Markov Game $\mathcal{M}^\phi$, and refer to $\mathcal{E}$ as an **oracle**. In the remainder of this paper, we slightly abuse the notation to also refer to algorithms implementing an oracle as $\mathcal{E}$.

**Definition 2.4.** Given a Stackelberg-Markov game $\mathcal{S}$ and a follower best-response oracle $\mathcal{E}$, a pair $(\phi^*, \pi^*)$ is a **Stackelberg-Markov** equilibrium if $\phi^*$ maximizes the principal's expected payoff under the condition that $\pi^*$ is drawn from $\mathcal{E}(\mathcal{M}^\phi)$; i.e. that $\phi^* \in \underset{\phi \in \Phi}{\arg\max} \underset{\pi \sim \mathcal{E}(\mathcal{M}^\phi)}{\mathbb{E}} [u_o(\phi, \pi)]$.

# 3 ANALYSIS OF PRIOR METHODS

In this section, we formalize AI policymaking as finding a Stackelberg-Markov equilibrium in order to perform a principled analysis of Zheng et al. (2022)'s AI Economist and its limitations   through an ablation of their policy generator inputs. We additionally ablate MetaGrad Yang et al. (2022), a related method that differs only in the optimization process of the economic policy generator. Detailed explanations of AI Economist and MetaGrad can be found in subsection A.1 and subsection A.2 respectively.

We now begin by describing our generalized framework, where we model simulated environments as a POMG. For each principal action, we draw follower equilibria for the induced POMG from an oracle, an abstraction that allows for a clear theoretical separation of principal and follower learning problems.

---

**Algorithm 1** Generalized framework for AI policymaking

---

1: Initialize Stackelberg-Markov Game $\mathcal{S} = (n, \Phi, \Pi, \mathbf{u})$.
2: Initialize Oracle $\mathcal{E}$.
3: (Optional) Pre-train Oracle $\mathcal{E}$ with $\mathcal{M}^\phi$, where $\phi$ are sampled randomly for pretraining $\mathcal{E}$.
4: **while** Stackelberg-Markov equilibrium not reached, **do**
5:     Principal commits to an action $\phi \in \Phi$.
6:     Principal action induces POMG $\mathcal{M}^\phi = \left(S, A^\phi, T^\phi, r^\phi, \Omega^\phi, \gamma, \mu_0^\phi\right)$.
7:     Compute follower best-response$\pi \sim \mathcal{E}(\mathcal{M}^\phi)$ using oracle.
8:     Evaluate principal payoff $u_0(\phi, \pi)$.
9: **end while**

---

Note that $\phi$ can parameterize economic policies (tax rates) directly, or an economic policy generator (weights of a function that produces tax rates). We use this framework to formalize Zheng et al. (2022)'s AI Economist. In their GTB environment, an RL planner uses an economic policy generator network to set tax rates for ten tax periods within each episode, aiming to maximize social welfare amongst the agents. Their training proceeds in three distinct phases: **1.** free-market agent training with no planner, **2.** agent training under a heavily entropy-regularized planner policy, effectively producing tax rates at random, and **3.** standard MARL learning for the planner and agents in tandem. In the following, we explain phases 1 & 2 as training the oracle in line 3 of Algorithm 1, and phase 3 of this training structure as an implementation of lines 4 through 7 of Algorithm 1.

**Multi-task oracle pretraining.** In phases 1 & 2, agents are pretrained to yield an approximate follower best-response oracle $\mathcal{E}$. The free market phase adapts agents to general game dynamics, while phase 2 is a form of multi-task pretraining: agents observe the current tax rates, and thus high principal entropy trains responses to a wide variety of actions randomly sampled from $\Phi$.

**Stackelberg-Markov Game.** In each episode of phase 3, a principal action $\phi \in \Phi$ induces a POMG: $\mathcal{M}^\phi = (S, A, T, r^\phi, \Omega, O, \gamma, \mu_0)$, $r^\phi(s, a) = \text{tax}(r^{\text{raw}}(s, a), R(\phi, s))$, where $r^{\text{raw}}: S \times A \mapsto \mathbb{R}^n$

is a base environment reward vector; $\text{tax} : \mathbb{R}^n \times \{x \in \mathbb{R} : 0 \le x \le 1\}^7 \mapsto \mathbb{R}^n$ computes and redistributes taxes according to a 7-bracketed tax rate; and $R : \Phi \times S \mapsto \{x \in \mathbb{R} : 0 \le x \le 1\}^7$ uses $\phi$ and current POMG state to determine applicable tax rates. Note that principal actions $\phi$ parametrize a function that maps POMG state to tax rates, but are not the tax rates themselves. The AI Economist generator network, mapping POMG states and network weights to tax rates, is not a Stackelberg-Markov principal, but is the tax rate function $R$.[2] In this case, $\Phi$ is the space of economic policy generator *weights*; one such set of weights $\phi$ is chosen at the start of each episode, and their method uses gradient-based optimization to find a $\phi^*$ that induces Stackelberg-Markov equilibrium.

MetaGrad, on the other hand, differs from AI Economist only in how the economic policy generator is trained. Instead of traditional reinforcement learning, meta-gradients are differentiated through the follower best-response computation to its effect on the principal payoff – conceptually similarly to Finn et al. (2017)'s MAML.

The presence of $S$ in the domain of $R$ raises the question of what state-space features are necessary for $R$. In GTB, the AI Economist policy generator network takes as input large POMG state space observations, including agent endowments, market behavior, and in some of their experiments even RGB world observations. We perform our MetaGrad ablation on their Escape Room (ER, further described in subsection A.2) environment, due to ER being their best-performing one. Here, the policy generator network takes as input the locations of all agents and most of their recent actions.

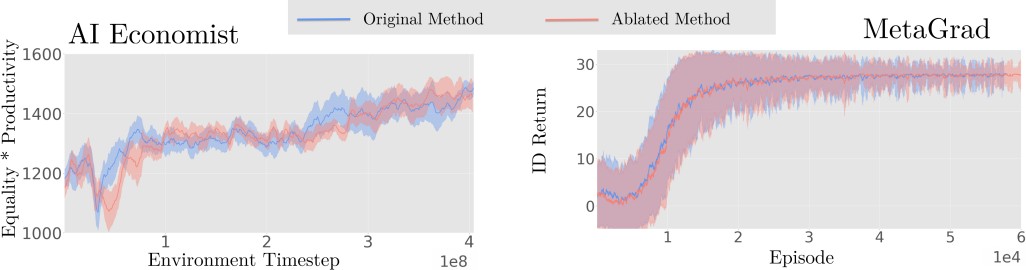

Figure 2: Ablation results for AI Economist on GTB and MetaGrad on Escape Room. Experiments were repeated over 3 and 10 seeds respectively.

Our ablations, shown in Figure 2, remove all observations from the AI Economist policy generator. Formally, this is equivalent to changing the domain of tax rate function $R : \Phi \times S$ to just $\Phi$. We keep the original architecture and input shape to avoid the need for a search over learning rates but overwrite observations to fixed vectors of ones. For MetaGrad, we removed the economic policy generator network entirely and learned the outputs directly as trainable parameters. With no network, their method reduces to Xu et al. (2018)'s meta-gradient reinforcement learning. Our ablated method uses a higher learning rate to account for the reduced number of parameters but otherwise uses identical hyperparameters to the original. By our formalization, this shows us that the principal action space has been unnecessarily complicated into a high-dimensional space of network weights. This points to a somewhat concerning problem with previous methods – that they appear to be overcomplicating the problem setting. Thus, a question is posed: can one dramatically improve the efficiency of learning by leveraging an altogether different approach?

## 4 EFFICIENT AI POLICYMAKING IN ECONOMIC SIMULATIONS

Motivated by our ablations, we propose a simpler method using Large Language Models as economic policymakers to directly output economic policies (e.g. tax rates). This also enables us to flexibly take in a wide variety of inputs beyond the current POMG state, of which in this paper we focus on contextualization and historical observations. We propose using the In-Context Learning (ICL) Dong et al. (2024) capabilities of the LLM to learn the optimal economic policy without updating any weights through leveraging contextualization and historical observation. We define

---

[2]More precisely, their economic policy generator takes as input an *observation* $o$ of $s$. This is equivalent if $o$ retains all state-space information required to compute tax rates, which we show holds in triviality.

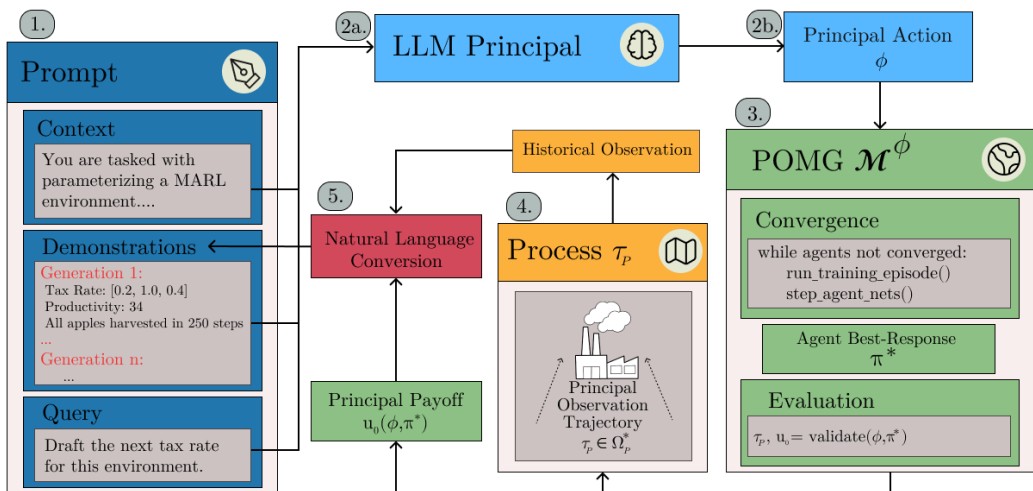

Figure 3: LLM Principal. At the beginning of each episode, **1.)** A prompt is built from three components: Context, an overview of the problem setting; Demonstrations, a documentation of previous principal actions and associated outcomes; and Query, a request for the next action $\phi$. **2.)** An LLM takes in this prompt and produces $\phi$. **3.)** $\phi$ induces a POMG $\mathcal{M}^\phi$. Agents train until reaching best-response policy $\pi^*$ under $\phi$, and we then evaluate $(\phi, \pi^*)$ within $\mathcal{M}^\phi$, yielding principal observation trajectory $\tau_P \in \Omega_P^*$ and principal payoff $u_0(\phi, \pi^*)$. **4.)** After evaluation, we process $\tau_P$, extracting historical data beyond $u_0(\phi, \pi^*)$. **5.)** Payoff and historical data are appended to the prompt history.

contextualizations as natural language descriptions of the problem setting to which our method is applied (Sodhani et al., 2021). Historical data we define by extending induced POMGs $\mathcal{M}^\phi$ in Stackelberg-Markov games to $(S, A^\phi, T^\phi, r^\phi, \Omega_F^\phi, \Omega_P, O_F^\phi, O_P, \gamma^\phi, \mu_0^\phi)$, augmented with a non-parameterizable principal observation space $\Omega_P$ and stochastic observation function $O_P$, distinct from the follower observation space and observation function $\Omega_F$ and $O_F$ respectively. At the end of an episode of $\mathcal{M}^\phi$, the principal POMG observation trajectory $\tau_P \in \Omega_P^* := \bigcup_{i \geq 0} \Omega_P^i$ is summarized into "historical observations" to inform the next choice of principal action $\phi'$. Note that this is in contrast to the AI Economist's use of historical observations to inform choices of $\phi$ via the loss and generator network optimization step. For more detail on the extended POMG, please refer to Appendix D, where we provide detailed groundwork to assist future work in incorporating a larger variety of information into automated policy-making methods, which adds further detail to our above extension.

Our LLM principal method, as illustrated in Figure 3, queries an LLM for an initial action choice conditioned on a contextualization of the problem setting and iteratively adds to this contextualization, appending successive action choices and their corresponding payoffs and historical observations. Additionally, to further increase the difficulty of our environments for GPT-4o mini, our strongest LLM, we occasionally include deliberate irrelevant information in contextualizations and historical observations received to test its ability to parse these effectively; the exact prompts we use are in Appendix H. Overall, the prompt structure for our method is as follows:

$$\textbf{<CONTEXT>}$$
$$\textbf{<ACTION}_1\textbf{><PAYOFF}_1\textbf{><HISTORICAL-OBSERVATION}_1\textbf{>}$$
$$\cdots$$
$$\textbf{<ACTION}_{n-1}\textbf{><PAYOFF}_{n-1}\textbf{><HISTORICAL-OBSERVATION}_{n-1}\textbf{>}$$
$$\textbf{<QUERY>}$$

Note the repetition of Action, Payoff, and Historical-Observation, in a similar format to CoT prompting Wei et al. (2022), but with historical observations rather than reasoning.

## 5 EXPERIMENTAL RESULTS

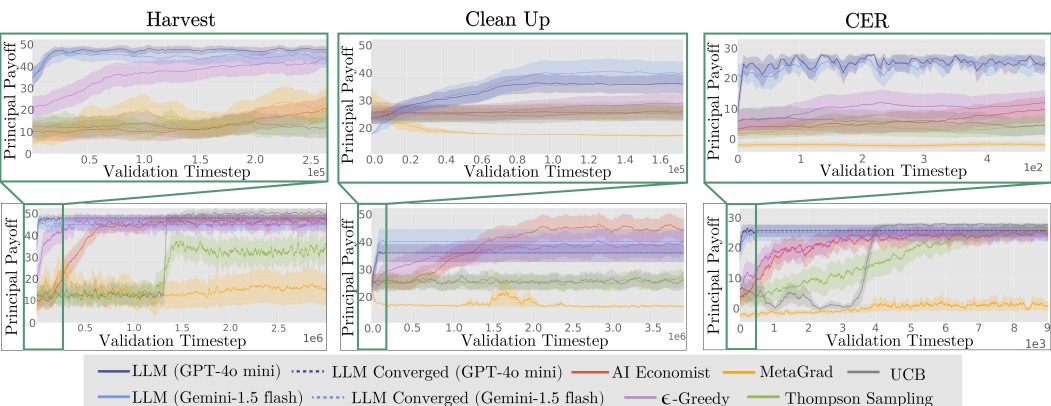

Figure 4: Performance comparison of different policymaker methods across the Harvest, Clean Up, and CER environments. Each plot displays the principal's reward over validation timesteps. Dashed lines represent principal payoff upon convergence to a policy. In addition to two frontier LLMs, we include RL-based methods of AI Economist, MetaGrad, and three bandit algorithms: UCB, Thompson sampling, and $\epsilon$-greedy. Both instantiations of the LLM principal consistently achieve higher sample efficiency than baselines across all environments. For each environment, we include a closer frame of reference for LLM performance in early timesteps (top) as well as the full run below. All methods were run on 10 seeds.

To demonstrate the effectiveness of our approach, we conduct experiments across three environments. Each environment was chosen to showcase social dilemmas where agents without external influence achieve low social welfare. We rigorously validate the following baselines: Zheng et al. (2022)'s AI Economist, Yang et al. (2022)'s MetaGrad, and three prevalent bandit algorithms: UCB (Auer et al., 2002), Thompson sampling (Thompson, 1933), and $\epsilon$-greedy (Sutton & Barto, 2018). Our LLM method is tested across two models: GPT-4o mini and Gemini-1.5 flash. In the following subsections, we evaluate our method against these baselines and explore the degree to which contextualizations affect performance. We had mixed success in ablating historical observations from our method; results and discussion are given in Appendix B. MetaGrad, and AI Economist were run with and without historical observations on all environments; we use each method's best performance in our main comparison in subsection 5.2, and ablate the effect of these in Appendix B. The LLM uses historical observations on all environments. subsection 5.3 ablates the effect of contextualization on the LLM's performance; we use contextualization in all final LLM experiments.

Consistent with standard economic models (Smith, 1776), our aim is to assess policymaking methods in environments where agents are assumed to be rational. Therefore, in Harvest and Clean Up we use pre-trained follower agents reset each principal action as implementations of oracles, as set out in Algorithm 1, and across all environments we allow agents to converge within $\mathcal{M}^{\phi}$ over multiple episodes for each action $\phi$. The latter serves to fine-tune agent policies and ensure the oracles do not violate Theorem 1 of Gerstgrasser & Parkes (2023). To fairly evaluate each method, we fix the number of agent convergence episodes performed for all methods on a per-environment basis. These numbers were chosen according to static tests as the minimal amount needed for agents to exhibit certain rational behaviors; namely, overharvesting under a free market environment in Harvest and Cleanup and successfully opening the door under the known global optimum incentives in CER. Principal actions $\phi$ are then evaluated against converged agent policies in validation episodes; as our environments are stochastic, some methods benefit from the evaluation of principal payoff averaged over multiple repeated episodes of $\mathcal{M}^{\phi}$. The number of validation episodes required is treated as a hyperparameter that we grid searched over for each method. All results are reported in terms of validation episode timesteps only, to avoid masking differences in validation episode requirements behind large numbers of convergence episodes[3].

---

[3]A choice that works to our method's disadvantage, since it requires the most validation episodes.

There is an exception, however, for MetaGrad. This method differentiates meta-gradients through agent policy update steps; scaling the inner loop of this procedure imposes considerable computational cost and memory burden, as well as vanishing gradient issues (Rajeswaran et al., 2019). MetaGrad therefore uses just one convergence episode across all environments and no agent resets. Extended graphs counting total convergence and validation environment timesteps showed the same overall trends and are given in Appendix A.3, where we provide a detailed discussion on the overall poor performance of MetaGrad in our environments.

## 5.1 ENVIRONMENTS

We employ three environments for our main experimental results: Harvest, Clean Up, and Contextual Escape Room. We focus on Stackelberg-Markov games with easily-interpretable principal action spaces. In the following subsections, we give an brief overview of our each environment – further detail can be found in subsection D.2.

**Commons Harvest Open (Harvest).** In Harvest (Agapiou et al., 2023), the environment contains several patches of apples, and agents $i \in [7]$ are given reward $+1$ for harvesting an apple. Harvested apples regrow stochastically at a rate proportional to the number of nearby unharvested apples – never regrowing once a patch has been exhausted. To sustainably accrue reward, agents must refrain from overharvesting and allow apples time to regrow. Without external influence, the environment succumbs to tragedy of the commons. In Harvest, the principal outputs a three-tiered tax rate to be applied to the reward signal of agents collecting apples, where the applicable tax rate is determined by the number of apples an agent has collected in the past 10 timesteps. Episodes are terminated after 1000 timesteps.

**Clean Up.** Clean Up (Jaques et al., 2019; Agapiou et al., 2023) also involves seven agents harvesting apples, but features a fundamentally different social dilemma. In this environment, a river builds pollution at a constant rate, and apples grow in a single, large patch at a rate inversely proportional to the river's pollution level. Note that apple regrowth is no longer affected by overharvesting. Agents can clean river pollution, but must leave the apple patch to do so. Harvesting apples yields a small intrinsic reward of $+0.1$ and cleaning a harsh penalty of $-1$. Under unmodified environmental rewards, agents remain in the apple patch without cleaning, even when pollution stops regrowth entirely. In this environment, the principal incentivizes three subsets of the agent action space, adjusting the reward signal for agents that harvest, clean, or do other actions. Episodes are terminated after 1000 timesteps.

**Contextual Escape Room (CER).** A $(n, m, L)$ Contextual Escape Room environment is an extension of Yang et al. (2020)'s Escape Room environment. This simple environment has $L + 2$ states, consisting of $L$ lever states, a door state, and a start state. Agents $i \in [n]$ begin an episode in the start state, and draw actions from a space equivalent to the state space – determining where they move to next. In each episode, one lever state $\ell$ at random is chosen to be "activated"; if $m$ agents are at lever $\ell$, the door "opens" and all agents at the door state receive a reward of $+10$. Moving to any other state, regardless of the door's status, incurs a penalty of $-1$, unless the door is closed and agents have not moved since the last step, in which case they receive 0. The social dilemma is that self-interested agents have no reason to incur a penalty for visiting lever states, and thus without incentivization the door is never opened. A $(n, m, L)$ CER environment has known global optimum of $10 \cdot (n - m) - m$ total agent reward per step. We employ a $(5, 2, 3)$ CER environment in which the principal observes the active lever and outputs five incentives to be added to the agent reward signal, corresponding to each state. Episodes are terminated after 5 timesteps.

## 5.2 COMPARISON OF METHODS

Here we present the direct comparison across all methods and baselines shown in Figure 9. Because of the high cost of LLM APIs for frontier models, we terminate LLM runs after several dozen steps of convergence – totaling 90 generated actions on Harvest, 60 on Clean Up, and 100 on CER. Note that the LLM and MetaGrad methods have continuous output spaces, whilst all others use discretized output spaces. We use a normal-inverse-gamma conjugate prior for Thompson Sampling initialised to high variance estimates, that in practice mean it begins by pulling a large proportion of arms, and initialise the UCB algorithm by one pull on each bandit arm following standard convention. The sharp increase in performance for UCB across several environments occurs at the end of this

exploration phase. To allow for a reasonable comparison, therefore, we decrease the number of arms for these two methods by using a coarser discretization. This decreases the cost of initial exploration, and we verified experimentally that optimal arms were not removed. Details of the discretization used along with all other hyperparameters can be found in subsection E.3; in all cases, the discretization rate we use is either equal to or more generous than that used by the original AI Economist method.

In Table 1, we provide an analysis of the results pictured in Figure 4. We show superior sample efficiency on all three environments, converging an order of magnitude faster than AI Economist on Harvest and Cleanup, and two orders of magnitude faster on CER, with similar or greater payoffs.

Table 1: Performance Comparison Across Environments. Mean and standard error principal payoff at convergence and timestep to reach convergence, across 10 seeds for each method. Convergence timestep was determined according to relative change within a rolling window – details and plots of individual runs can be found in Appendix I. We denote runs that do not converge or achieve trivial performance with DNC, and report their payoff at final timestep.

| Method | Harvest | | Clean Up | | CER | |
|---|---|---|---|---|---|---|
| | Timestep ($\times 10^3$) | Payoff | Timestep ($\times 10^3$) | Payoff | Timestep | Payoff |
| GPT-4o mini | $\mathbf{36 \pm 2}$ | $46.4 \pm 0.7$ | $90 \pm 5$ | $35.6 \pm 0.7$ | $\mathbf{32 \pm 7}$ | $25.3 \pm 1.7$ |
| Gemini-1.5 flash | $51 \pm 1$ | $44.4 \pm 2.4$ | $\mathbf{89 \pm 5}$ | $39.2 \pm 2.2$ | $34 \pm 5$ | $24.4 \pm 1.6$ |
| AI Economist | $984 \pm 108$ | $44.7 \pm 0.7$ | $2465 \pm 149$ | $\mathbf{43.9 \pm 1.0}$ | $5050 \pm 591$ | $23.5 \pm 0.5$ |
| $\epsilon$-greedy | $587 \pm 122$ | $44.0 \pm 0.4$ | $2003 \pm 307$ | $38.2 \pm 1.1$ | $4840 \pm 593$ | $24.0 \pm 0.2$ |
| UCB | $1574 \pm 22$ | $\mathbf{47.5 \pm 0.1}$ | DNC | $25.9 \pm 0.6$ | $4301 \pm 31$ | $\mathbf{26.4 \pm 0.1}$ |
| Thompson | $2195 \pm 104$ | $32.4 \pm 0.3$ | DNC | $26.6 \pm 0.5$ | $8126 \pm 196$ | $25.1 \pm 0.2$ |
| MetaGrad | DNC | $16.5 \pm 2.1$ | DNC | $16.4 \pm 0.2$ | DNC | $2.5 \pm 3.3$ |

**Harvest**: Both LLMs were significantly more sample efficient than baselines and reached similar final payoffs. GPT-4o mini converged to a payoff greater than all baselines except UCB, which was orders of magnitude less sample efficient. AI Economist notably used almost twice as many samples to converge as $\epsilon$-greedy, and both reached similar final payoffs. MetaGrad on average performed poorly; some runs reached similar payoffs to the LLMs and AI Economist, but others increased the first tax rate significantly, leading to agents not harvesting at all. The LLMs were less robust to noise than baselines, and we found in grid searches that they required three repeated validation episodes whilst all baselines performed best with just one.

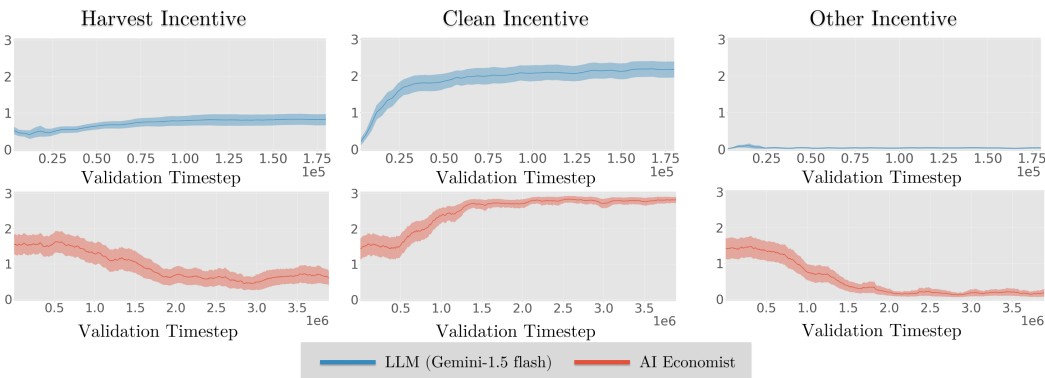

Figure 5: Incentives over time in the Clean Up environment for Gemini-1.5 flash (top) and AI Economist (bottom). From left to right, graphs correspond to the incentive for harvesting, cleaning, and other. Note the LLM x-axis is an order of magnitude smaller than the AI Economist x-axis.

**Clean Up**: This environment is more stochastic than Harvest, and grid searches showed all methods performed best with three validation episodes. Notably, AI Economist performed strongly here, reaching highest asymptotic performance. The main driving factor behind this is that the "other actions" incentive needs to be set close to zero for agents to act sensibly, leading to many of the bandit arms yielding low payoffs. AI Economist produces incentive sets using a separate action head for each incentive instead of considering an exponential number of bandit arms, and was there-

fore able to rapidly set this third incentive to zero independently of the other two. The LLMs converged to lower final payoffs than AI Economist, but once again required far fewer samples to converge. Gemini-1.5 flash reached the same final payoff as $\epsilon$-greedy, though significantly faster, and converged higher than all other baselines except AI Economist. To explain this disparity in final payoff, we note that AI Economist converged to a final set of incentives around $[0.5, 2.75, 0.05]$ ([harvest, clean, other]), and Gemini-1.5 close to this at approximately $[0.5, 2.0, 0.05]$, as shown in Figure 5. While on some seeds the LLM pushed the cleaning incentive higher, it failed to do so on others; responses indicated a tendency to avoid pushing incentives to their extremes, particularly in counter-intuitive cases where an incentivized action is not directly aligned with the action that yields principal payoff (e.g. increasing cleaning incentive to improve harvesting).

**CER**: Shorter episodes in CER reduce principal payoff stochasticity, and we found all methods performed best on one validation episode only. GPT-4o-mini again converges to a higher payoff than all baselines other than UCB and equal to that of Thompson Sampling, though both of these use a heavily discretized ($[0, 2.5, 5]$) range for each incentive and are several orders of magnitude less sample efficient. Gemini-1.5-flash observes similar effects to that of GPT-4o-mini. AI Economist performs very similarly to the much simpler $\epsilon$-greedy bandit, and both are again several orders of magnitude less sample efficient than the LLMs to reach similar final payoffs. MetaGrad once again performed poorly; we discuss this in detail in subsection A.3.

## 5.3 CONTEXTUALIZATION ABLATION

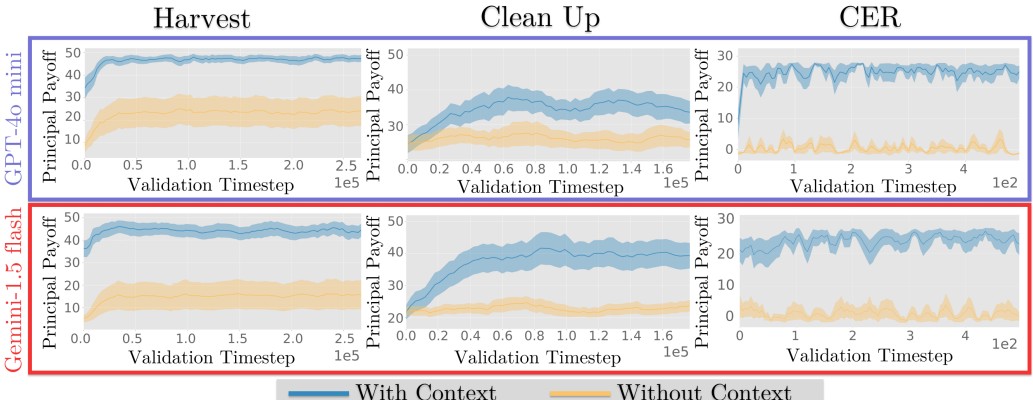

Figure 6: Effect of contextualization on LLM performance for each model tested, across all environments. Gemini-1.5 flash and GPT-4o mini are ablated on 10 seeds per environment. We compare to results reported in subsection 5.2, clearly demonstrating the advantage of leveraging pretraining.

In this section, we conduct an ablation on our method's performance with and without contextualization of the problem setting. In the contextualization case, the LLM received a detailed description of the environment and task at hand, including an explanation of what its outputs are used for. Without contextualization, the LLM is asked to produce outputs to maximize an unspecified function, relying purely on in-context learning. Results, shown in Figure 6, demonstrate that LLM performance significantly improves when contextualization is included - consistently across all environments and both models tested. These findings indicate that when properly contextualized, LLMs have the potential to leverage their extensive pre-trained knowledge to improve policymaking. As the scale of LLM pre-training continues to increase (Naveed et al., 2023), this opens up possibilities for more efficient AI-driven policymaking, where models can incorporate nuanced economic principles learned from diverse sources during pre-training into decisions. It is unclear how such contextualizations could be effectively utilized by baseline methods.

## 6 RELATED WORK

While the exploration of LLMs specifically concerning economic policymaking in MARL environments is limited beyond previously outlined methods, we now outline research within similar fields.

**Contextual Attention-based Representation Learning** (CARE) (Sodhani et al., 2021) identifies the difficulty of achieving strong multi-task performance for RL-based autonomous agents. Sodhani et al. (2021) attribute this to an inability to leverage auxiliary information between tasks and propose the usage of a pretrained LM to encode task descriptions, used to condition the policy of an RL agent. CARE employs multiple encoders, each focusing on a specific component of a task. While CARE achieved impressive results in several robotic manipulation tasks, using an end-to-end LLM approach allows our method to leverage more fine-grained auxiliary information without being restrained by a set number of encoders.

**Reward Design with Language Models** (Kwon et al., 2023) attempts to mitigate the human expense of reward design within RL. Rather than directly providing expert demonstrations or a specific reward function, Kwon et al. (2023) provide a natural language description of desired agent behavior to an LLM. The LLM is then tasked with evaluating agent behavior and subsequently producing a reward signal. Unlike our work, rewards are reactively assigned based on agent action.

**Simulating Human Behavior** aims to instantiate generative agents as realistic human proxies. One such approach (Park et al., 2023) does so in a sandbox environment, using extended LLMs with memory banks to store experiences unique to each agent. Vinitsky et al. (2023) introduces MARL environments with 'public sanctions', in which agents equipped with a novel architecture learn social norms and punish transgressors, facilitating agent cooperation in socially complex environments. Research towards this end indicates the possibility of future AI-policymaker tests that better evaluate real-world performance.

## 7 CONCLUSION

In this work, we standardize and evaluate existing methodologies in the field of AI-driven economic policy design. We further contribute a novel method, leveraging LLMs for a more generalizable and tractable approach that outperforms baselines . The relative infancy of this field engenders a lengthy research agenda; importantly, one such direction for future work is increasing the realism and complexity of the economic environments being tested. Although we show that prior methods are overly complex for existing environments and directly outputting a static tax rate is enough, in the real-world, a tax rate generator may be necessary and more complex environments should be developed to investigate this further.

### 7.1 LIMITATIONS

As simulated environments increase in complexity, prompts given to instantiations of LLM policymakers will presumably increase as well. This calls into question the scalability of prompt design, as well as the memory constraints imposed by the context windows of LLMs. Furthermore, we found in our experiments that LLMs can demonstrate a tendency to trap themselves in local minima, choosing to make small adjustments to past outputs rather than attempting a more dramatic shift.

### 7.2 ETHICS

Though we strongly believe that research in this field has the potential to benefit society broadly, we also realize that the development of these systems comes with risk. Though that this field is still years away from the technological possibility of real-world implementation, it is essential to remember that technological difficulty should not be considered the lone obstacle. Equally importantly, if not more so, these systems must be rigorously evaluated for any existent biases imparted by developers or training data. Furthermore, defining social welfare in the real world will be an involved task, and should likely be multifaceted, unlike our naive implementation. The continued development of this field should involve a diverse set of people from different backgrounds, professions, and viewpoints. Progress should be fully transparent, and made digestible to those outside of technical research. For this reason, we will be publishing an open-access website with a simplified summary of this work upon publication, as well as all source code.

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

## A  METHOD OVERVIEW

### A.1  AI ECONOMIST

AI Economist is an application of deep policy-based model-free RL to the economic policymaking problem: the economic policymaker is modelled as an additional agent in the environment, at every timestep receiving an observation, sampling an action from a policy network and receiving a reward. The policy network, serving as an economic policy generator, has a separate discrete action head for each tax bracket, each a categorical distribution over a discretized output range with an additional NO-OP action. The economic policy generator network outputs at timesteps corresponding to the start of a tax period are set as tax rates in the environment, and all other actions are masked to NO-OPs. In GTB, rewards are given for each episode timestep as the difference in social welfare between that timestep and the last; in our environments, since only one action is reinforced per episode, we give each a reward according to the total social welfare achieved in that entire episode.

### A.2  METAGRAD

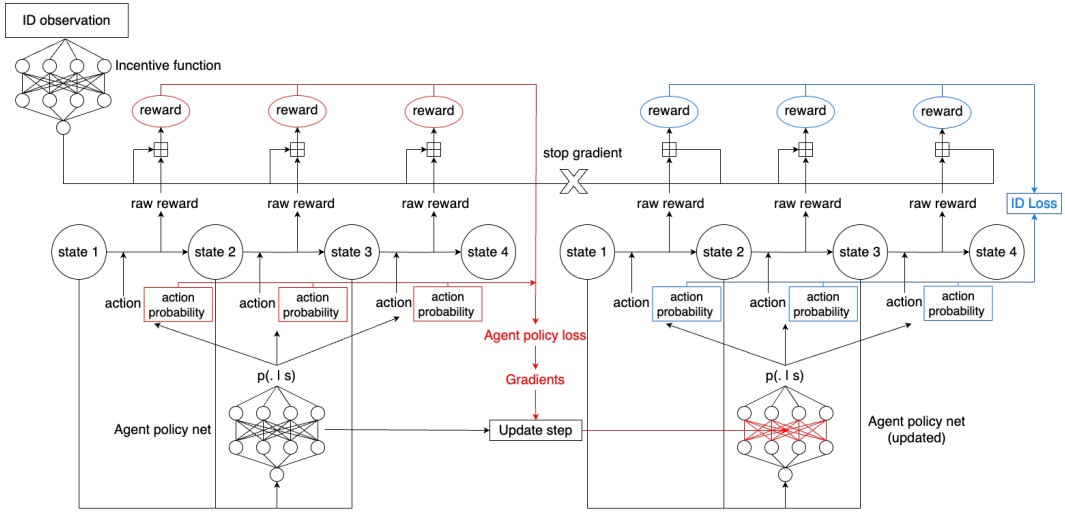

Figure 7: Visualization of gradient flows in MetaGrad.

Yang et al. (2022)'s MetaGrad method train a economic policy generator with meta-gradients flowed back through player agent parameter updates. Their method is conceptually similar to MAML (Finn et al., 2017) and we describe it here. A detailed flowchart to supplement our explanation is shown in Figure 7.

The environment we ablate their method on is named Escape Room; this was the basis for our Contextual Escape Room, and is the same but with just one lever state and no "activated levers". At each step, their principal, termed an *Incentive Designer (ID)*, observes both the positions of all agents and their most recent joint action and decides an incentive for visiting each state at that timestep. This incentive is decided using a forward pass through a neural network, termed the *incentive function*. When the agents subsequently act, their base environmental rewards have the correct incentives added – and crucially these modified rewards maintain gradient flow back to the incentive function's parameters. The player agents act for one episode of the game collecting rewards, and then form a normal RL loss from these rewards – for example, a policy gradient loss or a PPO (Schulman et al., 2017) loss through advantage calculations that maintain gradient flow. This loss is therefore still a differentiable function of the incentive function parameters. At this point, for agent parameters $\theta$, incentive function parameters $\eta$ and trajectory $\tau$, we have a loss $\mathcal{L}(\tau, \theta, \eta)$. We step the agent nets by SGD $\theta' \leftarrow \theta - \alpha \nabla \mathcal{L}(\tau, \theta, \eta)$ or more generally any update rule $\theta' \leftarrow \theta + f(\tau, \theta, \eta)$. These updated agent policy networks – with differentiable parameters $\theta'(\eta)$ – are used to collect a second trajectory $\tau'$, this time with no gradient flow back through rewards to the incentive function parameters. In this trajectory, the ID receives a reward at each timestep according

to any social welfare metric we wish for it to optimize, such as mean pre-incentive agent reward. We aim to maximize ID rewards in this validation trajectory, so form any loss that has gradients maximizing $J^{\text{ID}}(\eta) := \mathbb{E}_{\tau' \sim \pi_{\theta'(\eta)}} \left[ \sum_{t=0}^{T} \gamma^t r_t^{\text{ID}} \right]$, such as a policy gradient or PPO loss – noting this is slightly unusually for *ID* rewards on a trajectory sampled from *agent* policy. We step this loss flowing meta-gradients back to the incentive function's parameters $\eta$, through the agent parameter update and the effect the new agent parameters have on our validation loss $J^{\text{ID}}$.

## A.3 METAGRAD PERFORMANCE IN OUR ENVIRONMENTS

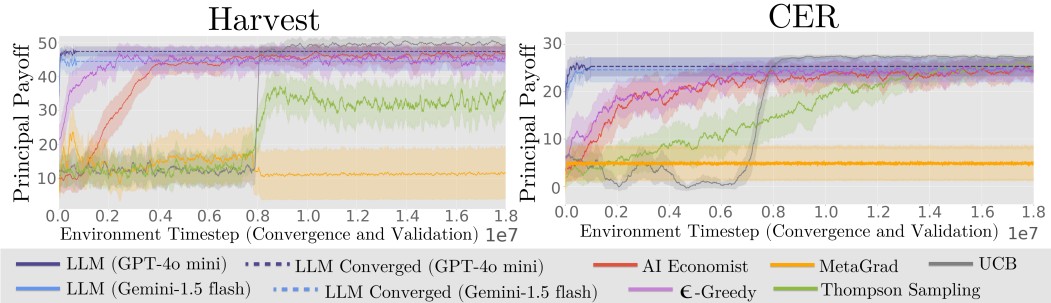

Figure 8: Copy of plots from Figure 4 counting total environment timesteps, including convergence and validation episodes, for Harvest and CER. For the latter, this involved running AID for 1000x more principal steps than all other baselines. Clean Up is not included as we were unable to get MetaGrad to work at all on this environment – we explain this below.

Here, we discuss MetaGrad's shortcomings on our test environments. Figure 8 shows copies of our main results for Harvest and CER counting all environment timesteps, to demonstrate our choice of using validation episode timesteps does not disguise significantly better MetaGrad performance.

In CER, even with the lever indicator fixed for all episodes instead of varying randomly, MetaGrad failed to successfully incentivize agents to open the door. Consistently, the incentive function increased both the correct lever incentive *and the door state incentive* to their largest allowable values. Clearly, the meta-gradients carry a consistently positive signal for the door state – and in fact we found that MetaGrad also fails on the original Escape Room environment without an incentivization cost loss as used in the original paper. We managed to get MetaGrad to work on fixed-indicator CER by adding an incentivization cost and carefully tuning its learning rate to catch the door state incentive increase – that has a gradient of subtly smaller magnitude than that of the correct lever incentive – but when lever randomization was reintroduced all incentivization cost learning rates we tried lead to all incentives going to zero before MetaGrad could learn to adapt to the changing indicator.

In Clean Up, all hyperparameter sets we considered failed completely. Sigmoid final layer activations mean the incentive function initially outputs values around 1.5 for each of the three incentives – including the third "other actions" incentive. Under most learning rates, this third incentive was decreased too slowly, leading to agents freezing completely in the initial phase of training. Once agents have frozen and no longer collect any reward, differentiating through agent policy network updates carries little information back to incentive function parameters, and incentives thereon only minorly oscillate with no distinguishable pattern. All learning rates we tried that were large enough to avoid this issue lead to highly unstable learning.

To verify whether the issue was our use of a policy gradient loss, or the fact that we have principals acting just once per episode, we tried allowing MetaGrad to act at every timestep, with a full critic network and a PPO validation episode loss. This was not successful either. Overall, the hyperparameters – specifically the principal learning rate – were very difficult to tune; perhaps we did not consider the right sets in our grid searches.

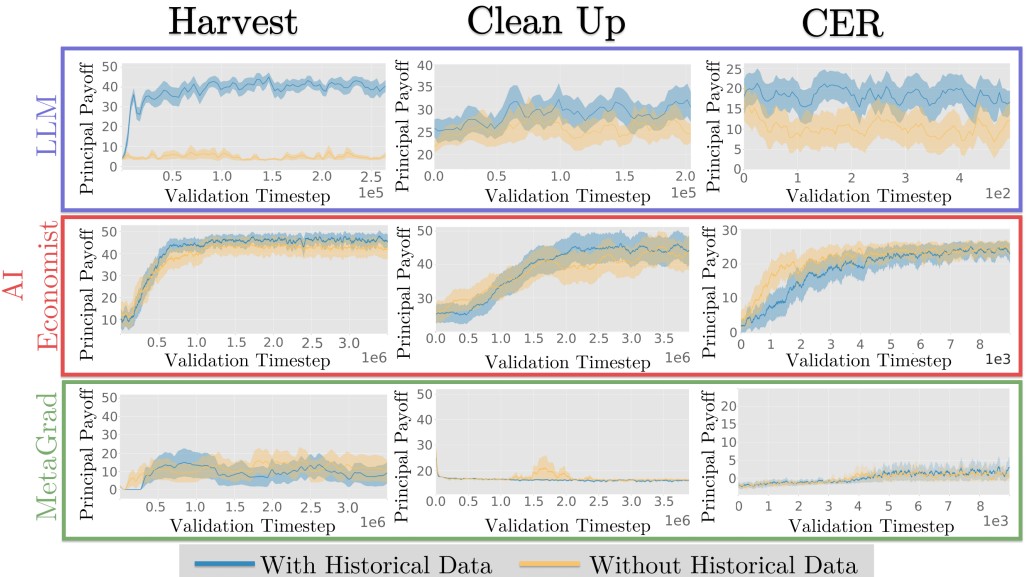

Figure 9: Legacy LLM Gemini-1.0 pro (top), AI Economist (middle), and MetaGrad (bottom) with and without historical observations on all environments. Methods with historical observations are denoted in blue, and methods without are denoted in yellow.

## B  HISTORICAL OBSERVATIONS

In Figure 9, we investigate the effect of historical observations on a legacy Gemini-1.0 pro LLM, AI Economist and MetaGrad performance across our three environments. For the LLM, these are given in natural language alongside randomly chosen pieces of irrelevant information – see our detailed prompt breakdown in Appendix H for examples. The LLM method without simply omits these from prompts. For AI Economist and MetaGrad, historical observations are the observations their method originally takes in; our ablated version maintains the same architecture and input shapes, but with fixed constant 1 observations.

We had mixed success in ablating historical observations from the frontier LLMs used in our main results, as these models were often able to perform strongly both with and without historical observations. As such, we use the weaker Gemini-1.0 pro model here instead. Other than the LLM model, all experiments use the hyperparameters and architectures used for our main results and described in Appendix E.

These results demonstrate that in our environments, as in GTB and Escape Room, AI Economist and MetaGrad are not able to make effective use of historical observations. However, for at least some models, our method demonstrates improvement when these are included. Firstly, this evidences that there exists useful information in these historical observations. Secondly, this suggests that as test environments are scaled in complexity commensurate with model capabilities, our method with frontier models may be able to leverage historical observations to further improve policymaking.

## C  IRRELEVANT INFORMATION EXPERIMENT

Historical information given to policymakers may at times be flawed or inaccurate; here, we demonstrate the capability of LLMs to parse potentially confusing information, irrelevant to the problem setting. In Figure 10, we see that this information has no detrimental effect on the performance of Gemini-1.5 flash in the Harvest environment. The experiment above was run for 90 tax rate generations, and the irrelevant sentences were added to each new set of tax rates, resultant payoffs, and (genuine) historical observations. Examples of these sentences include "George Washington knew better than to cut down an apple tree", "Six agents wore red, and one wore yellow", and "Apples are

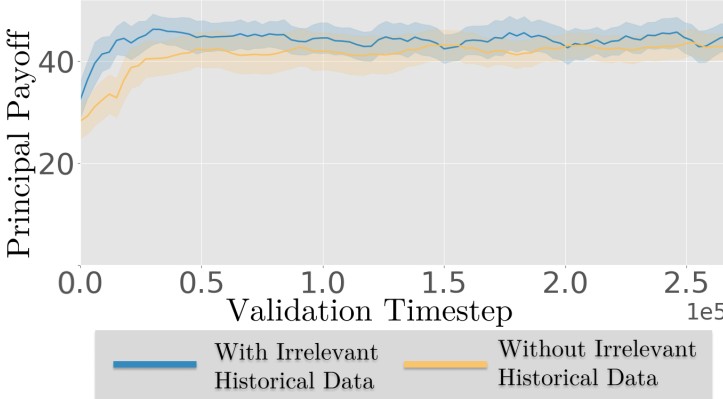

Figure 10: Comparison of Gemini-1.5 flash performance on Harvest with and without irrelevant information included in the prompt, evaluated over 10 seeds.

a member of the rose family, like pears and plums". We omit these sentences from the prompt of "Without Irrelevant Historical Data" and otherwise used identical prompts.

## D    HIERARCHICAL OBSERVATION STACKELBERG-MARKOV GAME

### D.1    DEFINITION

To formalize efficiently-informed economic policymaking, we extend Stackelberg-Markov games to allow for principal actions to be conditioned on observations. This is separate to the effects of state-space observations on the outputs of principal actions within POMGs, and we thus term these outer-level observations *macro*-observations. We use these to condition principal action choice on summarized information from the previous episodes POMGs and contextualizations of problem settings, though the macro-observation space can be richer.

**Definition D.1.** A **Hierarchical Observation Stackelberg-Markov Game** (HOSMG) $\mathcal{S} = (n, \mathcal{O}, \Phi, \psi, P, \Pi, \mathbf{u}, f)$ is a $n$-follower online Stackelberg-Markov Game, where $\mathcal{O}$ is the principal macro-observation space; $\Phi$ is the principal action space; $\psi : \mathcal{O} \mapsto \Phi$ is a principal policy; $P : \Phi \mapsto \mathcal{M}^\phi$ is a policy implementation map from principal actions $\phi \in \Phi$ to parameterized POMGs $\mathcal{M}^\phi = (S, A^\phi, T^\phi, r^\phi, \Omega_F^\phi, \Omega_P, O_F^\phi, O_P, \gamma^\phi, \mu_0^\phi)$ augmented with a non-parameterizable principal observation space $\Omega_P$ and stochastic observation function $O_P$, distinct from the follower observation space and observation function $\Omega_F$ and $O_F$ respectively; $\Pi = \bigtimes_{i \in [n]} \Pi_i$ is the followers' joint policy space; $\mathbf{u} = (u_i)_{i=0}^n$ holds the payoff functions $u_i : \Phi \times \Pi \mapsto \mathbb{R}$; $f : \Omega_P^* \mapsto \mathcal{O}$ maps POMG principal observation trajectories in $\Omega_P^* := \bigcup_{i \geq 0} \Omega_P^i$ to macro-observations.

Principal actions $\phi$ are now produced by a policy $\psi$ and macro-observations, and we augment induced POMGs $\mathcal{M}^\phi$ with an additional observation space $\Omega_P$ and stochastic observation function $O_P$ for the principal. At the end of each episode, a trajectory of principal POMG observations $o_t^P \in \Omega_P$ in $\mathcal{M}^\phi$ can be mapped by $f$ to a macro-observation in $\mathcal{O}$, the domain of $\psi$ – and thus the next choice of $\phi$ can be formally conditioned on a previous episode's events. Worked examples of this formalization are given in subsection D.2.

For an RL-based approach such as AI Economist, $\psi$ is the optimizer step used to determine the next set of weights $\phi$. The HOSMG formalization explicitly clarifies our view that these weights $\phi$ should be chosen informed by macro-observations drawn from a rich space $\mathcal{O}$ including, but not limited to, summarized state-space observations from the previous episode's POMG and contextualizing game descriptions. Sodhani et al. (2021) discuss the latter in detail, but, overall, it is not clear how one would go about incorporating large amounts of data in potentially many modalities efficiently into gradient-based optimizer steps. We take a different approach entirely, but tackling this problem directly represents a potential avenue for future work.

## D.2 Environment Formalizations

Here, we formalize each of our environments as HOSMGs.

**Harvest.** We augment this environment with a principal equipped with action space $\Phi = [0, 1]^3$. Principal actions $\phi = (R_1, R_2, R_3)$ are interpreted as three-tiered tax rates, corresponding to three tax brackets $\{[\tau_b, \tau_{b+1}] : b = 1, 2\} \cup \{[\tau_3, \infty]\}$. Actions $\phi$ parameterize $\mathcal{M}^\phi = (S, A, T, r^\phi, \Omega_A, \Omega_P, O_A, O_P, \gamma, \mu_0)$, $r_i^\phi(s, a) = r_i^{\text{raw}}(s, a) - \text{tax}_i(s, a) + \frac{1}{n} \sum_{j=1}^n \text{tax}_j(s, a)$, where $\text{tax}_i$ computes an amount of tax owed at each episode timestep by agent $i$, and we redistribute the total tax collected evenly. We tax the raw environmental reward collected by each agent at that timestep at a rate determined by their cumulative[4] raw reward in the past $H$ steps: $\text{tax}_{i,t} = \alpha \cdot R_{\text{bracket}\left(\sum_{k=t-H}^t r_{i,k}^{\text{raw}}\right)} \cdot r_{i,t}^{\text{raw}}$, $\text{bracket}(x) = \sum_{b=1}^3 b \cdot \mathbf{1}_{\tau_b \leq x < \tau_{b+1}}$, where $\alpha$ is a scalar multiplier allowing rewards to be made negative. Principal payoff is calculated as the mean end-of-episode cumulative endowment over all agents, equivalently pre-tax or post-tax. The principal POMG observation function $O_P : S \times A \mapsto \Delta(\Omega_P)$ deterministically maps states to the number of apples the environmental state they represent contains; $f : \Omega_P^* \mapsto \mathcal{O}$ for the LLM maps to the number of apples remaining at the end of a game, and for AI Economist and MetaGrad to trajectories of remaining apples.

**Clean Up.** We augment this environment following Yang et al. (2022), with a principal equipped with action space $\Phi = [0, 2]^3$. Principal actions $\phi = (I_1, I_2, I_3)$ are interpreted as incentives to be added at each timestep to the raw environmental rewards of agents that respectively: harvest an apple, perform a cleaning action, perform any other action. Actions $\phi$ parameterize $\mathcal{M}^\phi = (S, A, T, r^\phi, \Omega_A, \Omega_P, O_A, O_P, \gamma, \mu_0)$, where $r_i^\phi(s, a) = r_i^{\text{raw}}(s, a) + I_1 \cdot \mathbf{1}_{\{a_i = \text{"harvest"}\}} + I_2 \cdot \mathbf{1}_{\{a_i = \text{"clean"}\}} + I_3 \cdot \mathbf{1}_{\{a_i = \text{"else"}\}}$. Principal payoff is the mean number of apples harvested during an episode. The principal POMG observation function $O_P : S \times A \mapsto \Delta(\Omega_P)$ deterministically maps state-action pairs $(s, a)$ to observations containing: the number of apples in the environmental state $s$ represents; the number of harvesting actions $\sum_{i=1}^n \mathbf{1}_{\{a_i = \text{"harvest"}\}}$ in joint action $a$; the number of cleaning actions $\sum_{i=1}^n \mathbf{1}_{\{a_i = \text{"clean"}\}}$ in joint action $a$. For the LLM, the function $f : \Omega_P^* \mapsto \mathcal{O}$ then maps the two former to the total number of apples that regrew in an episode and the latter to the total number of cleaning actions that occurred in an episode. For AI Economist and MetaGrad, $f$ maps to downsampled trajectories of these two quantities.

**CER.** We augment this environment with a principal that incentivizes the action of visiting each state. Principal actions $\phi \in M_{L \times (L+2)}([0, 5])$ are matrices used to produce incentive sets $(I_1, ..., I_{L+2})$ that are added to raw environmental rewards as in Clean Up above. The POMG state space $S$ contains an indicator of which lever $\ell$ is activated, and the set of $L + 2$ incentives to be used for a reward $r^\phi(s, a)$ is extracted from $\phi$ as $e_\ell^T \phi$. In practice, given this lever indicator is fixed throughout each episode, we can equivalently query the principal for the relevant set of incentives only by providing the lever incentive to be used in advance. Principal payoff is calculated as the agents' cumulative joint raw reward collected each episode, averaged over each timestep. The principal POMG observation function $O_P(s, a) = (s_{\text{door}}, a)$ deterministically keeps joint actions and whether the door is open or closed in a state $s$; $f : \Omega_P^* \mapsto \mathcal{O}$ then summarizes the door's status, the active lever, and the average number of agents moving to each state per timestep of an episode.

# E EXPERIMENTAL SETUP

## E.1 AGENT ARCHITECTURES AND PRETRAINING

In Commons Harvest Open and Clean up, we train player agents with parameter sharing, PPO Schulman et al. (2017) and GAE Schulman et al. (2016). Agent actor and critic networks share three convolutional layers followed by one fully-connected layer, with separate single linear layer heads. Flattened output from the convolutional layers is concatenated to a one-hot player indicator and the vector of current tax rates / incentives before the actor and critic heads. Agents are pretrained and reset to their initial parameters at each principal step, after which they are allowed to converge

---

[4]Formally, agent cumulative endowments are folded into the POMG state space $S$, so that $\text{tax}_{i,t} := \text{tax}_i(s_t, a_t)$ does not violate the Markov property.

for 5 episodes in Commons Harvest Open and 20 in Clean Up. These values were determined by static tests and fixed prior to running any of our main experiments. We pretrain agents in two phases:

1. The whole agent network is trained to convergence in a free-market / incentive-free environment, after which convolutional layers are frozen.

2. Actor and critic heads are trained on 27 uniformly chosen tax rates / incentive sets in simultaneous parallel games.

In Contextual Escape Room, agents have 3-layer MLP policy networks, trained on a simple reward-to-go form policy gradient loss with an entropy regularization term and no parameter sharing. Agents are not pretrained, and train from initialization for 2000 episodes at each principal action.

### E.2    PRINCIPAL ARCHITECTURES

**AI Economist.** For all environments, we use actor-critic MLP networks with two hidden layers, trained with PPO and GAE. The principal economic policy generator has a discrete action head for each tax bracket / incentive. In Commons Harvest Open and Clean Up we discretize these into 21 points with an additional no-op action, and in Contextual Escape Room only 6 for this method to remain tractable.

**MetaGrad.** For all environments, the MetaGrad incentive function is a two layer MLP with a sigmoid final layer activation, scaled to the desired incentive range. Meta-gradients are flowed back from a policy gradient loss in validation episodes, using agent log-probabilities and principal rewards calculated as the difference in social welfare between each timestep. A running mean baseline is used to reduce variance.

**Bandit algorithms.** We discretize the tax rates and incentives to produce bandit arms. In CER, we use a separate set of arms for each lever indicator, and coarsen discretization to account for the increase in number of arms.

### E.3    HYPERPARAMETERS

We used Cartesian grid searches to tune hyperparameters for each baseline. Agent hyperparameters, including learning rate, were chosen according to a grid search on static tests and fixed for all methods as part of our oracle abstraction – though we grid searched over agent learning rate for MetaGrad to attempt to improve its performance as much as possible.  The hyperparameter ranges used for tuning are as follows. Each hyperparameter set was evaluated on 3 seeds, except $\epsilon$-greedy which was tested on 8.

- **LLMs**:
    - Harvest:
        * Validation episodes: [1, 3]
        * Temperature: [0.01, 0.5, 0.9]
    - Clean Up:
        * Validation episodes: [1, 3, 5]
        * Temperature: [0.01, 0.5, 0.9]
    - CER:
        * Validation episodes: [1, 3]
        * Temperature: [0.01, 0.5, 0.9]
- **AI Economist**:
    - Harvest:
        * Validation episodes: [1, 3]
        * Principal LR: [5e-5, 1e-4, 5e-4, 1e-3]
        * Principal hidden layer dimension: [128, 256]
        * Principal $c_{\text{entropy}}$: [0.05, 0.2]
    - Clean Up:
        * Validation episodes: [1, 3, 5]

* Principal LR: [5e-5, 1e-4, 5e-4]
            * Principal hidden layer dimension: [128, 256]
            * Principal $c_{\text{entropy}}$: [0.05, 0.2]
        – CER:
            * Validation episodes: [1, 3]
            * Principal LR: [5e-5, 1e-4, 5e-4, 1e-3]]
            * Principal hidden layer dimension: [64, 128]
            * Principal $c_{\text{entropy}}$: [0.2, 0.488]
    * **MetaGrad**:
        – Harvest and Clean Up:
            * Principal LR: [1e-4, 2e-4, 3e-4, 7e-4]
            * Agent LR: [1e-5,1e-4,1e-3]
            * MetaGrad hidden layer dimension: [128, 256]
        – CER:
            * Principal LR: [1e-5, 1e-4, 3e-4, 7e-4, 1e-3]
            * Agent LR: [1e-5,1e-4,1e-3]
            * MetaGrad hidden layer dimension: [64, 128]
    * **$\epsilon$-greedy**:
        – Harvest and CER:
            * Validation episodes: [1, 3]
            * $\epsilon$: [0.1, 0.2]
        – Clean Up:
            * Validation episodes: [1, 3, 5]
            * $\epsilon$: [0.1, 0.2]
    * **UCB**:
        – Harvest and CER:
            * Validation episodes: [1, 3]
            * $c_{\text{UCB}}$: [0.2, 0.5, 1]
        – Clean Up:
            * Validation episodes: [1, 3, 5]
            * $c_{\text{UCB}}$: [0.2, 0.5, 1]
    * **Thompson Sampling**:
        – Harvest and CER:
            * Validation episodes: [1, 3]
        – Clean Up:
            * Validation episodes: [1, 3, 5]

Final hyparameters used are given in Table 2 and Table 3.

### E.4 COMMONS HARVEST OPEN TAX MULTIPLIER

Our implementation of taxation in the Commons Harvest Open environment uses a fixed scalar multiplier $\alpha$ on each agent's due tax before taxation and redistribution. Agent $i$'s final reward post-tax, writing $b_i$ as shorthand for the tax bracket agent $i$ falls into, is:

$$r_{i,t} = r_{i,t}^{\text{raw}} - \alpha \cdot R_{b_i} \cdot r_{i,t}^{\text{raw}} + \frac{1}{n} \sum_{j=1}^{n} \alpha \cdot R_{b_j} \cdot r_{i,t}^{\text{raw}}$$

The motivation for this scaling is that, due to tax redistribution and our environment containing only 7 player agents, even under maximal taxation $[1, 1, 1]$ agents heavily overharvesting still receive a reward of $+\frac{1}{7}$ for collecting an apple. To allow the principals a more expressive range of reward modifiers, we scale collected taxes to allow for effective rewards to be made negative. We fixed this

| Parameter | Harvest | Clean Up | CER |
|---|---|---|---|
| Agent LR | $3 \times 10^{-4}$ | $3 \times 10^{-4}$ | $1 \times 10^{-3}$ |
| Agent LR (MetaGrad) | $1 \times 10^{-3}$ | $1 \times 10^{-3}$ | $1 \times 10^{-3}$ |
| Convergence episodes | 5 | 20 | 500 |
| Minibatch size | 128 | 128 | - |
| $\gamma$ | 0.998 | 0.998 | 0.99 |
| $\lambda_{\text{GAE}}$ | 0.98 | 0.98 | - |
| PPO update epochs | 4 | 4 | - |
| PPO clip coef | 0.2 | 0.2 | - |
| PPO value clip coef | 0.2 | 0.2 | - |
| $c_{\text{value}}$ | 0.5 | 0.5 | - |
| Agent $c_{\text{entropy}}$ | 0.025 | 0.025 | 0.166 |
| Gradient clipping norm | 0.5 | 0.5 | - |
| Adam $\epsilon$ | $1 \times 10^{-5}$ | $1 \times 10^{-5}$ | $1 \times 10^{-5}$ |
| $\epsilon$-greedy epsilon | 0.1 | 0.2 | 0.1 |
| $c_{\text{UCB}}$ | 0.2 | 0.2 | 0.2 |
| AI Economist $c_{\text{entropy}}$ | 0.2 | 0.2 | 0.488 |
| AI Economist LR | $5 \times 10^{-4}$ | $1 \times 10^{-4}$ | $5 \times 10^{-4}$ |
| AI Economist hidden dim | 128 | 128 | 128 |
| AI Economist hidden layers | 2 | 2 | 2 |
| MetaGrad LR | $1 \times 10^{-4}$ | $1 \times 10^{-4}$ | $3 \times 10^{-4}$ |
| MetaGrad hidden dim | 256 | 256 | 64 |
| MetaGrad hidden layers | 2 | 2 | 2 |
| LLM temperature | 0.01 | 0.01 | 0.01 |
| Thompson NIG priors | $\mu = 0, \nu = 0.05, \alpha = 1, \beta = 25$ | | |

Table 2: Shared and method-specific hyperparameters.

| Parameter | LLM | AI Econ | MetaGrad | $\epsilon$-greedy | UCB | Thompson |
|---|---|---|---|---|---|---|
| Harvest discretization | - | $[0, 1, 0.05]$ | - | $[0, 1, 0.05]$ | $[0, 1, 0.1]$ | $[0, 1, 0.1]$ |
| Clean Up discretization | - | $[0, 3, 0.15]$ | - | $[0, 3, 0.15]$ | $[0, 3, 0.3]$ | $[0, 3, 0.3]$ |
| CER discretization | - | $[0, 5, 1]$ | - | $[0, 5, 1]$ | $[0, 5, 2.5]$ | $[0, 5, 2.5]$ |
| Harvest validation episodes | 3 | 1 | 1 | 1 | 1 | 1 |
| Clean Up validation episodes | 3 | 3 | 3 | 3 | 3 | 3 |
| CER validation episodes | 1 | 1 | 1 | 1 | 1 | 1 |

Table 3: Number of validation episodes and discretization used [start, end, step] for each method.

scaling factor to $4$ prior to running any experiments, according to the rough guide that a tax rate of $[0, 1, 1]$ should penalize an agent overharvesting with a $-1$ penalty. Here, we say overharvesting is occurring when around half of the agents fall in one of the two later tax brackets and have just harvested, in which case the reward for one of these agents under $\alpha = 4$ would be:

$$r_{i,t} = 1 - 4 \cdot (1 - \text{proportion agents overharvesting}) \approx 1 - \frac{4}{2} = -1$$

# F    COMPUTATIONAL RESOURCES

Experiments were primarily run using RTX 4090 GPUs sourced from RunPod. A small subset of experiments were run using a Titan Xp GPU and an RTX 3090 GPU. Throughout our research, we spent roughly $4,000 USD on RunPod, correlating to approximately 2,500 GPU hours, and $100 USD between the OpenAI and Gemini APIs.

## G IMAGES OF THE ENVIRONMENTS

Images of our environments are shown in Figure 11, Figure 12 and Figure 13.

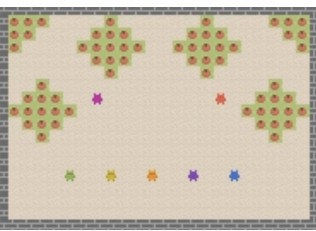

Figure 11: Commons Harvest Open. Seven agents populate an environment with several patches of apples. Apples can regrow unless a patch is fully harvested, and no apples remain.

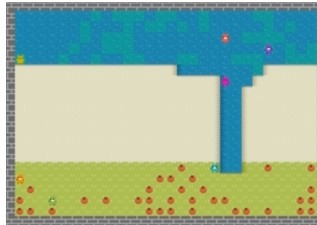

Figure 12: Clean Up. Seven agents populate an environment where a beach separates a large apple patch from a river that builds pollution (depicted as teal tiles).

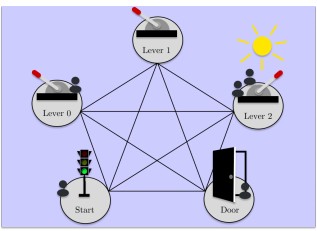

Figure 13: Rough depiction of Contextual Escape Room. Five agents populate an environment with five states: three lever states, a start state, and a door state. One lever is randomly activated each episode. If two or more agents go to the active lever state, the door opens.

## H PROMPTING

This section gives examples of prompt-response pairs across each model and each of our environments. For readability, we mark the initial context with blue, the history with green, and the query with brown. Experiments ablating contextualization use the same prompt for both GPT-4o mini and Gemini-1.5 flash .

### H.1 GPT-4O MINI

#### H.1.1 COMMONS HARVEST OPEN

**Prompt:**

You are overseeing agents in a MARL environment. These agents populate an environment that contains apples, and are rewarded when they harvest an apple. However, if all of the apples are harvested before they can regrow, the environment will be barren for the rest of the episode and the agents will be unable to increase their return. Without any tax rates, the agents overharvest, leaving none to regrow and getting a low mean return. You are charged with creating tax rates that will affect the reward signal for an agent harvesting an apple. Each 'taxed' reward will be redistributed evenly amongst all of the agents. Your goal is to maximize the mean return of the agents. The following history of your previous attempts tells you the attempt number, the three-tiered tax rate you previously generated for that attempt, and the approximate mean income that the tax rate resulted in, as well as other data that might be pertinent.

Generation 1: [0.1, 0.3, 0.5] ->mean apples: about 50. Six agents wore red, and one wore yellow.

Generation 2: [0.3, 0.5, 0.7] ->mean apples: about 9. Under this tax rate, apples remained unharvested at the end of the episode.

Generation 3: [0.2, 0.4, 0.6] ->mean apples: about 39.

Generation 4: [0.15, 0.35, 0.55] ->mean apples: about 49. Six agents wore red, and one wore yellow.

Generation 5: [0.12, 0.32, 0.52] ->mean apples: about 35. Apples are a member of the rose family, like pears and plums.

Now you will generate the next tax rate to be implemented in the environment, with the goal of maximizing the mean return of the agents. Please provide this tax rate in the form $ [tax1, tax2, tax3] $ where each tax rate must be a floating point value between 0 and 1 inclusive. Do not forget the brackets, the commas, and the dollar signs. Before you generate the tax rate, please explain your rationale behind choosing it (so the tax rate should be the last thing that you generate).

**Response:**

*Rationale*: *The goal is to maximize the mean return of the agents. We need to find a balance between encouraging agents to harvest apples and ensuring that some apples remain for regrowth.*

*Observations from previous attempts:*

- **Generation 1**: Tax rates of [0.1, 0.3, 0.5] yielded a mean of about 50 apples, which indicates that this was relatively effective for a low taxation system.
- **Generation 2**: Higher tax rates of [0.3, 0.5, 0.7] led to a drastic drop in mean apples to about 9, suggesting over-taxation discouraged harvesting.
- **Generation 3**: A moderate tax rate of [0.2, 0.4, 0.6] resulted in about 39 apples, indicating some balance between harvesting and regrowth.
- **Generation 4**: Tax rates of [0.15, 0.35, 0.55] performed well, yielding a mean of 49 apples, showing this balance worked effectively.

*Strategy:*

*We need to refine the tax rates based on these observations. Given that lower tax rates yield better results, but we need to maintain sustainability, the following steps should be taken:*

- A slight increase from Generation 1 will incentivize sustainable harvesting while encouraging long-term growth.
- We propose to test slightly lower taxes than Generation 5 but higher than Generation 2 for balance.

*Proposed Tax Rate:*

*$ [0.1, 0.3, 0.5] $*

### H.1.2 CLEAN UP

**Prompt:**

You are overseeing agents in a MARL environment. These agents populate an environment that contains apples, and receive a reward of 0.1 when they harvest an apple. There is also a river in the environment that builds pollution at a steady rate. The apples can regrow, but do so more slowly if the river is polluted. You will produce 3 modifiers to be implemented in an episode of this environment. Agents that harvest an apple will see a reward signal of (0.1 + modifier1), agents that clean the pollution will see a reward signal of (-1 + modifier2), and agents that do some other action will see a reward signal of (0 + modifier3). Your goal is to maximize the mean number of apples harvested.

Below, you are given the history of your previous attempts.

Generation 1: [2.0, 1.0, 0.0] - Agents were able to clean roughly 101 times. As a result, roughly 1 apple grew back. Agents harvested about 17 apples each. Apples are a member of the rose family, like pears and plums.

Generation 2: [2.0, 0.5, 0.0] - Agents were able to clean roughly 99 times. As a result, roughly 3 apples grew back. Agents harvested about 17 apples each. Happy birthday!

Generation 3: [2.5, 1.5, 0.0] - Agents were able to clean roughly 150 times. As a result, roughly 36 apples grew back. Agents harvested about 22 apples each.

Now you will generate the next set of modifiers to be implemented in the environment. Please provide this set of modifiers in the form $ [modifier1, modifier2, modifier3] $ where each modifier must be a floating point value that is nonnegative and not greater than 3. Do not forget the brackets,

the commas, and the dollar signs. Before you generate the modifiers, please explain your rationale behind choosing it (so the modifiers should be the last thing that you generate).

**Response:**

*Rationale*: To maximize the mean number of apples harvested, we need to consider the impact of each modifier on the agents' behavior and the overall environment.

*Observations from previous attempts:*

- **Generation 1**: The set of modifiers [2.0, 1.0, 0.0] led to agents cleaning roughly 101 times, resulting in only 1 apple regrowing and about 17 apples harvested per agent. This indicates that while cleaning was incentivized, apple regrowth was minimal.
- **Generation 2**: The modifiers [2.0, 0.5, 0.0] showed similar harvesting results (about 17 apples each) but increased the number of apples regrown to 3, suggesting a slight improvement.
- **Generation 3**: Using [2.5, 1.5, 0.0], agents cleaned more frequently (150 times), which significantly improved regrowth to 36 apples and increased the mean apples harvested to 22 per agent.

*Strategy:*

*Based on these observations, we need to maintain a balance that encourages both harvesting and pollution cleaning. The following adjustments are proposed:*

- *\*\*Modifier 1 (Reward for harvesting apples)\*\*: Since the reward in Generation 3 encouraged increased apple harvesting without overwhelming the agents, we will retain this value at 2.5 to maintain strong incentives.*
- *\*\*Modifier 2 (Reward for cleaning pollution)\*\*: The increased value in Generation 3 led to more frequent cleaning and better regrowth. We will maintain this value at 1.5 to ensure that agents are motivated to clean the pollution efficiently.*
- *\*\*Modifier 3 (Reward for other actions)\*\*: Keeping this value at 0.0 will discourage agents from engaging in non-productive actions, keeping their focus on harvesting and cleaning.*

*Proposed Modifiers:*

*$ [2.5, 1.5, 0.0] $*

### H.1.3 CONTEXTUAL ESCAPE ROOM

**Prompt:**

You are overseeing agents within a MARL environment. There are 5 states in this environment – lever 1, lever 2, lever 3, door, and start. Agents are already given environmental rewards. You will be providing 5 modifiers, between 0 and 5, to the predefined episode rewards – for each lever, the start, and the door. Your goal is to maximize the total environmental rewards collected by the agents. You do not know anything at all about the rules of the game. Lever 1 is painted green. A history of your previous attempts tells you the attempt number, the set of five incentives you previously generated for that episode, and the outcome of that episode. Here is your history of attempts:

Episode 1: you chose incentives [2.0, 5.0, 1.0, 1.0, 1.0]. The light was on above lever 2. The door remained closed. 0.0 agents went to lever 0, 5.0 agents went to lever 1, 0.0 agents went to lever 2, 0.0 agents went to the door state, and 0.0 agents went to the start. The agents received a total reward per step of -1.0.

Episode 2: you chose incentives [5.0, 1.0, 0.0, 0.0, 1.0]. Lever 0 is made of iron. The light was on above lever 0. The door remained closed. 2.2 agents went to lever 0, 0.0 agents went to lever 1, 0.0 agents went to lever 2, 2.8 agents went to the door state, and 0.0 agents went to the start. The agents received a total reward per step of 25.8.

Episode 3: you chose incentives [1.0, 1.0, 5.0, 0.0, 0.0]. A lever is a beam connected to the ground by a hinge or pivot, called a fulcrum. The light was on above lever 2. The door remained closed. 0.0

agents went to lever 0, 0.0 agents went to lever 1, 2.0 agents went to lever 2, 3.0 agents went to the door state, and 0.0 agents went to the start. The agents received a total reward per step of 28.0.

Episode 4: you chose incentives [1.0, 5.0, 0.5, 1.0, 0.0]. Lever 0 was photographed for a magazine about levers. The light was on above lever 1. The door remained closed. 0.0 agents went to lever 0, 2.2 agents went to lever 1, 0.0 agents went to lever 2, 2.8 agents went to the door state, and 0.0 agents went to the start. The agents received a total reward per step of 25.8.

We will now start a new episode. Please remember that the five incentives you generate should be in the form $ [lever 0 incentive, lever 1 incentive, lever 2 incentive, door state incentive, start state incentive] $ where each incentive must be a floating point value between 0 and 5 inclusive. Do not forget the brackets, the commas, and the dollar signs. Before you generate the next incentives, please explain the rationale behind your decision. A light is on above lever 2.

**Response:**

**Rationale:** *To determine the best set of incentives for the upcoming episode, we need to analyze the previous episodes and the outcomes based on the incentives provided and the states that were active.*

**Analysis of Previous Episodes:**

- **Episode 1**: *Incentives were [2.0, 5.0, 1.0, 1.0, 1.0]. The light was on above lever 2, but no agents went to lever 2. Instead, 5 agents went to lever 1. The total reward was -1.0, indicating that the incentives did not effectively guide agents to the desired state.*
- **Episode 2**: *Incentives were [5.0, 1.0, 0.0, 0.0, 1.0]. The light was on above lever 0, and 2.2 agents went to lever 0, resulting in a high reward of 25.8. This suggests that the incentive for lever 0 was effective.*
- **Episode 3**: *Incentives were [1.0, 1.0, 5.0, 0.0, 0.0]. The light was on above lever 2, and 2.0 agents went to lever 2, leading to a reward of 28.0. The high incentive for lever 2 likely attracted agents effectively.*
- **Episode 4**: *Incentives were [1.0, 5.0, 0.5, 1.0, 0.0]. The light was on above lever 1, and 2.2 agents went to lever 1, resulting in a reward of 25.8. The incentive for lever 1 was effective.*

**Current Episode Context:** *The light is on above lever 2. Based on the previous episodes, we know that when the light is on above a lever, it is crucial to provide a strong incentive for that lever to attract agents.*

**Rationale for New Incentives:**

- **Lever 0**: *Given that it was effective in Episode 2, we can keep a moderate incentive (around 1.0) to maintain some interest.*
- **Lever 1**: *This lever has shown effectiveness in Episode 4, so we can keep a moderate incentive (around 1.0) as well.*
- **Lever 2**: *Since the light is on above lever 2, we should provide a high incentive (around 5.0) to attract agents to this lever.*
- **Door State**: *The door has not been a focus in previous episodes, so we can keep this incentive low (around 0.0).*
- **Start State**: *This state has not been effective in previous episodes, so we can also keep this incentive low (around 0.0).*

**Proposed Incentives:** *Based on this analysis, the proposed incentives for the new episode are:*

*$ [1.0, 1.0, 5.0, 0.0, 0.0] $*

*This configuration aims to maximize the total environmental rewards by focusing on the active lever (lever 2) while maintaining some interest in the other states.*

## H.2 GEMINI-1.5 FLASH

### H.2.1 HARVEST

**Prompt:**

You are overseeing agents in a MARL environment. These agents populate an environment that contains apples, and are rewarded when they harvest an apple. You are charged with creating tax rates that will affect the reward signal for an agent harvesting an apple. Each 'taxed' reward will be redistributed evenly amongst all of the agents. Your goal is to maximize the mean return of the agents. Below, you are given the history of your previous attempts.

 Generation 1: [0.25, 0.3, 0.35] - mean apples: about 46.

Generation 2: [0.4, 0.35, 0.25] - mean apples: about 6. Under this tax rate, apples remained unharvested at the end of the episode. Apples are a member of the rose family, like pears and plums.

Now you will generate the next set of three tax rates to be implemented in the environment, with the goal of maximizing the mean return of the agents. The three tax rates you generate should be preceded by the phrase FINAL RATES where each tax rate must be a floating point value between 0 and 1 inclusive. Follow this format: FINAL RATES: [tax1, tax2, tax3]. Before you generate the tax rates, please explain your rationale behind choosing it (so the tax rates should be the last thing that you generate).

**Response:**

*Rationale for Tax Rate Selection*:

*The goal is to maximize the mean return of the agents, which translates to maximizing the number of apples harvested. We need to find a balance between incentivizing individual agents to harvest apples and ensuring a fair distribution of rewards.*

*Analysis of Previous Generations:*

- **Generation 1:**
    - *A tax rate of [0.25, 0.3, 0.35] resulted in a mean of about 46 apples harvested. This suggests that the tax rates were not too high, allowing for a decent amount of harvesting.*
- **Generation 2:**
    - *A tax rate of [0.4, 0.35, 0.25] led to a drastic decrease in the mean apples harvested (only 6). This indicates that the tax rates were too high, discouraging agents from harvesting.*

*Key Observations:*

- ***High tax rates discourage harvesting**: The drastic drop in apples harvested in Generation 2 suggests that agents were not motivated to collect apples when facing high tax rates.*
- ***Tax redistribution is important**: The redistribution of taxed rewards ensures that all agents benefit from the collective effort, even if they don't directly harvest an apple.*

*Strategy for Generation 3:*

*Based on the above observations, we need to find a balance between incentivizing individual harvesting and ensuring a fair distribution of rewards. We will adjust the tax rates from Generation 1, aiming for a slightly lower tax rate to encourage more harvesting.*

*FINAL RATES:*

*$ [0.2, 0.25, 0.3] $*

### H.2.2 CLEAN UP

**Prompt:**

You are overseeing 7 agents in a MARL environment. These agents populate an environment that contains apples, and receive a reward of 0.1 when they harvest apple. There is also a river in the environment that builds pollution at a steady rate. The apples can regrow, but do so more slowly if

the river is polluted. Agents can clean the pollution, but receive a negative reward (-1) for doing so. You will produce 3 modifiers to be implemented in an episode of this environment. These modifiers will be applied to the reward signal for an agent based on the action that it takes, such that agents that harvest an apple will see a reward signal of (0.1 + modifier1), agents that clean the pollution will see a reward signal of (-1 + modifier2), and agents that do some other action will see a reward signal of (0 + modifier3). Your goal is to maximize the mean number of apples harvested.

The following history of your previous attempts tells you the attempt number, the three modifiers you previously generated for that attempt, and the mean income that the modifiers resulted in, as well as other information that may be pertinent.

Generation 1: [0.5, 0.0, 0.0] - Agents were able to clean roughly 22 times, and roughly 0 apples grew back. Agents harvested about 17 apples each. Six agents wore red, and one wore yellow.

Generation 2: [0.7, 0.5, 0.0] - Agents were able to clean roughly 64 times, and roughly 0 apples grew back. Agents harvested about 17 apples each. Six agents wore red, and one wore yellow.

Generation 3: [0.3, 1.0, 0.0] - Agents were able to clean roughly 108 times, and roughly 15 apples grew back. Agents harvested about 19 apples each. Six agents wore red, and one wore yellow.

Now you will generate the next set of modifiers to be implemented in the environment. Please provide this set of modifiers in the form $ [modifier1, modifier2, modifier3] $ where each modifier must be a floating point value that is nonnegative and not greater than 3. Do not forget the brackets, the commas, and the dollar signs. Before you generate the modifiers, please explain your rationale behind choosing it (so the modifiers should be the last thing that you generate).

**Response:**

*Rationale for Modifier Selection*:

*The goal is to maximize the mean number of apples harvested. We need to balance the following factors:*

- *Apple Growth**: The environment's apple regrowth rate is heavily influenced by pollution. Cleaning pollution is crucial for sustainable apple harvesting.*
- *Cleaning Cost**: Cleaning pollution incurs a negative reward (-1), which discourages agents from cleaning.*
- *Harvesting Reward**: The base reward for harvesting is 0.1, which needs to be incentivized to encourage agents to harvest.*

*Analysis of Previous Attempts:*

- **Generation 1:**
  - *A high modifier for harvesting (0.5) led to a decent harvest, but the lack of cleaning resulted in no apple regrowth.*
- **Generation 2:**
  - *Increasing the harvesting modifier further (0.7) didn't significantly improve the harvest, and the lack of cleaning continued to hinder apple regrowth.*
- **Generation 3:**
  - *A higher modifier for cleaning (1.0) significantly increased cleaning attempts, leading to some apple regrowth. However, the lower harvesting modifier (0.3) resulted in a lower overall harvest.*

*Proposed Strategy:*

*We need to find a balance between incentivizing cleaning and harvesting. We'll focus on the following:*

- *Increase Cleaning Incentive**: We'll increase the cleaning modifier to encourage more cleaning, leading to better apple regrowth.*

- **Maintain Harvesting Incentive**: *We'll keep a moderate harvesting modifier to ensure agents are still motivated to harvest.*
- **Explore Other Actions**: *We'll introduce a small modifier for other actions to see if it encourages agents to explore alternative strategies.*

***Proposed Modifiers:***

*$ [0.4, 1.5, 0.1] $*

### H.2.3 CONTEXTUAL ESCAPE ROOM

### H.3 CONTEXTUAL ESCAPE ROOM

**Prompt:**

You are overseeing agents within a MARL environment. There are 5 states in this environment – three levers, door, and start. You will be providing 5 incentives, between 0 and 5, corresponding to each lever, the start, and the door.

Note that agents are already given a +10 incentive for being at the door – but only if it opens. At the beginning of the episode, the door is closed and gives no rewards to the agents. Your goal is to maximize the total environmental rewards collected by the agents. The only thing in the environment that changes between episodes is the location of a light. Below, you are provided with a history of your attempts:

Attempt 1: the light was on above lever 0. You chose incentives [4.5, 1.5, 1.5, 2.5, 3.5]. No states were highly incentivized, comparatively. 1.0 agents went to lever0, 0.0 to lever1, 0.0 to lever2, 3.0 to the door, and 1.0 to the start. The door remained closed. The agents received a total reward of -0.8 per step.

Attempt 2: the light was on above lever 0. You chose incentives [5.0, 1.5, 1.5, 4.0, 2.5]. No states were highly incentivized, comparatively. 1.0 agents went to lever0, 0.0 to lever1, 0.0 to lever2, 4.0 to the door, and 0.0 to the start. The door remained closed. The agents received a total reward of -1.0 per step.

Attempt 3: the light was on above lever 1. You chose incentives [1.5, 5.0, 1.5, 4.5, 2.0]. No states were highly incentivized, comparatively. 0.0 agents went to lever0, 1.0 to lever1, 0.0 to lever2, 4.0 to the door, and 0.0 to the start. The door remained closed. The agents received a total reward of -1.0 per step.

We will now start a new independent episode. The incentives you generate must be floating point values between 0 and 5 inclusive. Follow this format: FINAL INCENTIVES: [lever0, lever1, lever2, door, start]. Explain your rationale without repeating yourself. Then, provide the incentives in the specified format. In your explanation, provide a detailed analysis of the history. There is now a light above lever 1. Remember that you are generating a set of five incentives. Each needs to be a floating point number between 0

### H.4 GEMINI-1.0 PRO

In the following section, the historical data that was withheld from the LLM in the historical data experiment B, will be underlined in red.

### H.4.1 HARVEST

**Prompt:**

You are overseeing agents in a MARL environment. These agents populate an environment that contains apples they can collect. However, if all of the apples are harvested before they can regrow, the environment will be barren for the rest of the episode. Without any tax rates, the agents overharvest, leaving none to regrow and getting a low mean return. You are charged with creating tax rates that will affect the reward signal for an agent harvesting an apple. Your goal is to maximize the mean return of the agents. Below, you are given the history of your previous attempts.

 Generation 1: [0.5, 0.7, 0.9] - mean apples: 3.

Under this tax rate, apples remained unharvested at the end of the episode.

 Generation 2: [0.4, 0.6, 0.8] - mean apples: 3.

Under this tax rate, apples remained unharvested at the end of the episode.

 Generation 3: [0.3, 0.5, 0.7] - mean apples: 6.

Under this tax rate, apples remained unharvested at the end of the episode.

 Generation 4: [0.2, 0.4, 0.6] - mean apples: 36. Generation 5: [0.25, 0.3, 0.35] - mean apples: 47. Generation 6: [0.5, 0.5, 0.5] - mean apples: 4.

Under this tax rate, apples remained unharvested at the end of the episode.

Now you will generate the next tax rate to be implemented in the environment, with the goal of maximizing the mean return of the agents. Please provide this tax rate in the form $ [tax1, tax2, tax3] $ where each tax rate must be a floating point value between 0 and 1 inclusive. Do not forget the brackets, the commas, and the dollar signs. Before you generate the tax rate, please explain your rationale behind choosing it (so the tax rate should be the last thing that you generate).

**Response:**

***Rationale***: *Based on the previous attempts, it can be observed that tax rates that are too high (Generation 1, 2, 3, 6) result in apples remaining unharvested, while tax rates that are too low (Generation 4) lead to overharvesting. Generation 5 seems to have found a good balance, with a mean return of 47 apples.*

*To further improve upon Generation 5, we can try slightly increasing the tax rates to encourage more conservative harvesting. Therefore, the next tax rate to be implemented is:*

*$ [0.3, 0.35, 0.4] $*

### H.4.2 CLEAN UP

**Prompt:**

You are overseeing 7 agents in a MARL environment. These agents populate an environment that contains apples, and receive a positive reward when they gather these apples. There is also a river in the environment that builds pollution at a steady rate. You will produce 3 incentives to be implemented in an episode of this environment, such that agents that harvest an apple will receive an additional reward of incentive1, agents that clean the river will receive a reward of incentive2, and agents that do some other action will receive a reward of incentive3. Your goal is to maximize agent harvesting.

The following history of your previous attempts tells you the attempt number, the incentives you previously generated for that attempt, and the level of agent harvesting that the incentives resulted in.

 Generation 1: [2.5, 1.0, 0.0] -

With the amount of cleaning agents did (about 120 cleaning actions) and the resulting pollution level in this episode, 143 apples regrew.

Agents harvested approximately 38 apples.

Now you will generate the next set of incentives to be implemented in the environment. Please provide this set of incentives in the form $ [incentive1, incentive2, incentive3] $ where each incentive must be a floating point value that is nonnegative and not greater than 3. Do not forget the brackets, the commas, and the dollar signs. Before you generate the incentives, please explain your rationale behind choosing it (so the incentives should be the last thing that you generate).

**Response:**

***Rationale***: *In the previous attempt, the agents harvested a relatively low number of apples (38) despite a high number of cleaning actions (120). This suggests that the incentive for cleaning (1.0)*

*was too low to encourage agents to prioritize cleaning over harvesting. To address this, we will increase the incentive for cleaning to 2.0. We will also slightly increase the incentive for harvesting to 2.75 to further encourage agents to focus on harvesting. Finally, we will set the incentive for other actions to 0.0 to discourage agents from engaging in non-productive behaviors.*

**Proposed Incentives:**

*$ [2.75, 2.0, 0.0] $*

### H.4.3 CONTEXTUAL ESCAPE ROOM

**Prompt:**

You are overseeing agents within a MARL environment called Escape Room. There are 5 states in this environment – three levers, door, and start. You will be providing 5 incentives, between 0 and 5, corresponding to each lever, the start, and the door. Your goal is to maximize agent performance. The only thing in the environment that changes between episodes is the location of a light. Below, you are provided with a history of your attempts:

Attempt 1: the light was on above lever 1. You chose incentives [2.5, 5.0, 2.5, 5.0, 1.0].

The door incentive and base environment reward for the door drew most agents strongly to the door - but it was closed.

The agents received a total reward of -1.0 per step.

We will now start a new independent episode. The incentives you generate must be floating point values between 0 and 5 inclusive. Follow this format: FINAL INCENTIVES: [lever0, lever1, lever2, door, start]. Explain your rationale without repeating yourself. Then, provide the incentives in the specified format. In your explanation, provide a detailed analysis of the history. There is now a light above lever 1. Remember that you are generating a set of five incentives. Each needs to be a floating point number between 0 and 5 and your final answer should follow this structure, REPLACING the words with your incentives: FINAL INCENTIVES: [lever0, lever1, lever2, door, start]

**Response:**

*Analysis of History:*

*In Attempt 1, the high incentive for the door (5.0) and the base environment reward for opening the door drew agents strongly to the door, even though it was closed. This resulted in a low total reward of -1.0 per step.*

*Rationale for New Incentives:*

*To improve agent performance, we need to:*

- *Reduce the incentive for the door to prevent agents from wasting time trying to open it.*
- *Increase the incentive for lever 1, as the light above it indicates that it is likely to be the correct lever to pull.*
- *Maintain a moderate incentive for the start state to encourage agents to explore the environment.*

**FINAL INCENTIVES:**

*$ [2.0, 4.0, 2.0, 1.0, 2.5] $*

- *Lever 0: 2.0 - Reduced incentive to prevent agents from getting stuck on this lever.*
- *Lever 1: 4.0 - Increased incentive due to the light above it.*
- *Lever 2: 2.0 - Maintained moderate incentive for exploration.*
- *Door: 1.0 - Reduced incentive to discourage agents from trying to open it.*
- *Start: 2.5 - Maintained moderate incentive for exploration.*

## H.5 PROMPTS WITHOUT CONTEXT

### H.5.1 HARVEST

**Prompt:**

You are attempting to maximize an unknown function $w(x, y, z)$. You will provide a set of inputs $(x, y, z)$, each between 0 and 1, to the function and observe the output. Below, you are given the history of your previous attempts.

 Generation 1: [0.5, 0.2, 0.8] - $\sim$ 4. 

Please provide the next set of three inputs. Each input must be a floating point value between 0 and 1 inclusive. Before you generate the inputs, please explain your rationale behind choosing them (so the inputs should be the last thing that you generate).

**Response:**

*Rationale*:

*Since we only have one data point, it's difficult to determine the function's behavior. A reasonable approach is to explore different regions of the input space. We'll try increasing one variable at a time, while keeping the others constant. This will help us understand the function's sensitivity to each input.*

*Proposed Inputs:*

*[0.5, 0.2, 0.9]*

### H.5.2 CLEAN UP

**Prompt:**

You are attempting to maximize an unknown function $w(x, y, z)$. You will provide inputs $(x, y, z)$, each between 0 and 3, to the function and observe the output. Below, you are given the history of your previous attempts.

 Generation 1: [1.5, 1.5, 1.5] - $\sim$ 21. 

Generation 2: [1.0, 1.5, 1.5] - $\sim$ 22. 

Please provide the next set of inputs in the form $[x, y, z]$ where each number must be a floating point value between 0 and 3 inclusive. Do not forget the brackets, the commas, and most importantly, the dollar signs. Before you generate the inputs, please explain your rationale behind choosing it (so the inputs should be the last thing that you generate).

**Response:**

*Rationale*:

*To maximize the unknown function $w(x, y, z)$, we need to analyze the previous attempts. In Generation 1, the input [1.5, 1.5, 1.5] yielded an output of approximately 21. In Generation 2, the input [1.0, 1.5, 1.5] resulted in a slightly higher output of approximately 22. This suggests that reducing the x value from 1.5 to 1.0 while keeping y and z constant at 1.5 may have led to a better output.*

*Given this trend, it seems that the function may be sensitive to changes in the x value, and that keeping y and z constant at 1.5 might be beneficial. To explore further, I will try a value for x that is slightly lower than 1.0, while keeping y and z the same. This could help identify if the function continues to increase as we lower x. Therefore, I will choose x = 0.5, y = 1.5, and z = 1.5 for the next input.*

*This choice allows us to explore the function's behavior in the lower range of x while maintaining the previously successful values for y and z.*

*Proposed Inputs:*

*$ [0.5, 1.5, 1.5] $*

### H.5.3 Contextual Escape Room

**Prompt:**

You are attempting to maximize an unknown function $u(A, v, w, x, y, z)$. Input A will be given to you and in response, you will provide inputs $[v, w, x, y, z]$, where each input must be a floating point value between 0 and 5 inclusive. You do not know anything at all about the rules of the function. Below, you are provided with a history of your attempts:

Attempt 1: A = 2. You chose inputs: $[2.5, 2.5, 2.5, 2.5, 2.5]$. $u(2, 2.5, 2.5, 2.5, 2.5, 2.5)$ = -1.6.

Attempt 2: A = 0. You chose inputs: $[0.0, 5.0, 0.0, 5.0, 0.0]$. $u(0, 0.0, 5.0, 0.0, 5.0, 0.0)$ = -1.0.

We will now try a new set of inputs. Please remember that the five inputs you generate should be in the form \$ [v, w, x, y, z] \$ where each input must be a floating point value between 0 and 5 inclusive. Do not forget the brackets, the commas, and the dollar signs. Before you generate the next inputs, please explain the rationale behind your decision. Attempt 3: A = 1.

**Response:**

*Rationale:*

*In the previous attempts, we observed that the function u(A, v, w, x, y, z) returned negative values for the inputs we provided. In Attempt 1, with A = 2 and inputs [2.5, 2.5, 2.5, 2.5, 2.5], the output was -1.6. In Attempt 2, with A = 0 and inputs [0.0, 5.0, 0.0, 5.0, 0.0], the output was -1.0. This suggests that the function may be sensitive to the values of v, w, x, y, and z, and that certain combinations may yield better results than others.*

*Given that the input values in Attempt 2 resulted in a less negative outcome than those in Attempt 1, it may be beneficial to experiment with a mix of values around these ranges to further explore how the function responds. Therefore, for Attempt 3, I will try varying the inputs slightly around those used in Attempt 2, while introducing some intermediate values to understand the impact on the output.*

*Proposed Inputs:*

*\$ [2.0, 3.0, 1.5, 4.0, 2.0] \$*

# I CONVERGENCE TIMESTEP CALCULATIONS

Timestep to convergence for each run is calculated by applying a rolling window to smoothed runs, breaking when maximum relative change within the window is $< 5\%$. On Harvest and CER, this window is 400 timesteps long, except for the much shorter LLM runs where it is 10 timesteps long; on Clean Up double these. These values were chosen by gradually increasing the window length until the mean timestep to convergence best reflected trends shown by the averaged runs for each method. Since there is room for ambiguity in that choice, all convergence timesteps determined by this process for each individual run are shown below with their calculated mean (thick black vertical line). DNC runs are not shown.

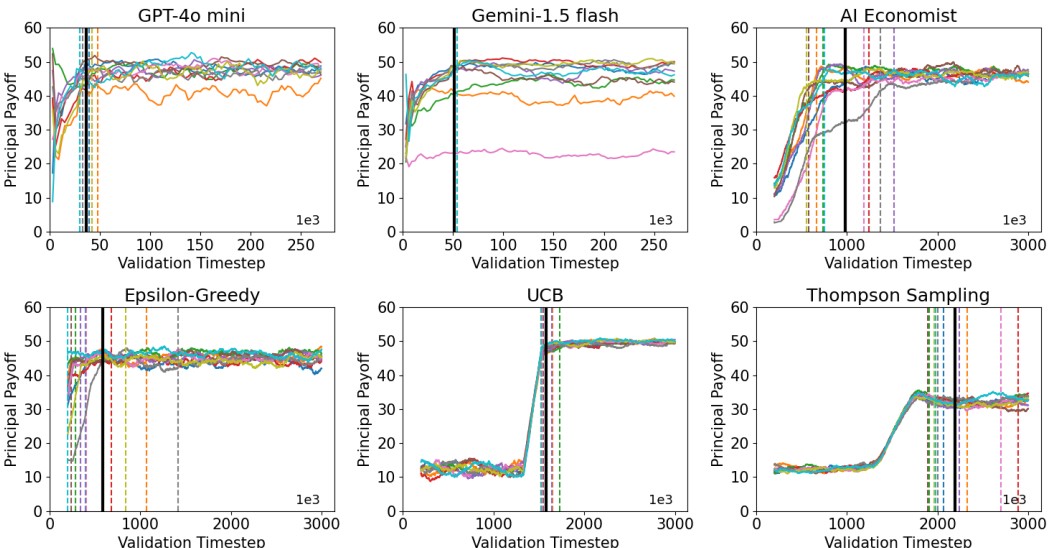

Figure 14: Convergence timesteps for Harvest.

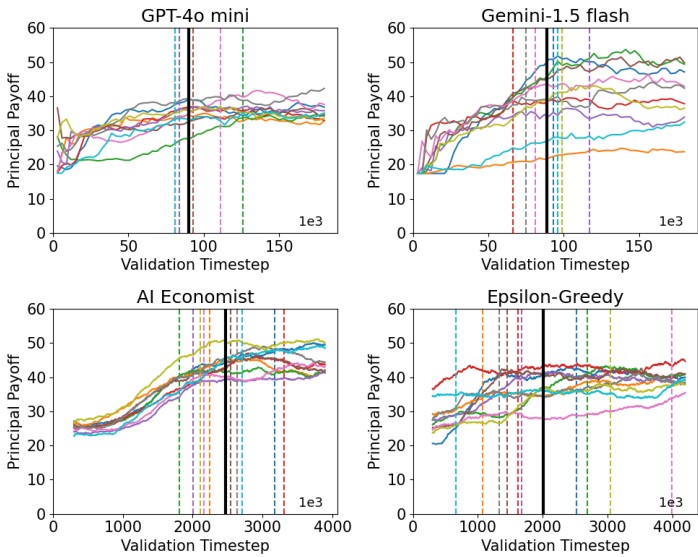

Figure 15: Convergence timesteps for Clean Up

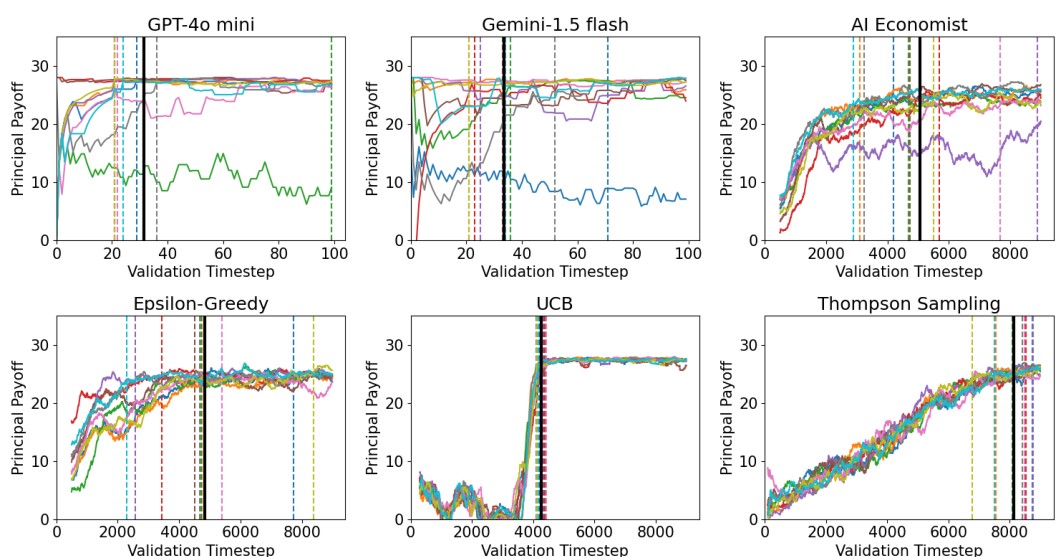

Figure 16: Convergence timesteps for CER

