# OpenReview forum: "Large Legislative Models: Towards Efficient AI Policymaking in Economic Simulations"
_ICLR.cc/2025/Conference — Submitted to ICLR 2025_

### Official Review · Reviewer_SeUt · 2024-10-19

**Soundness:** 2
**Presentation:** 1
**Contribution:** 2
**Rating:** 3
**Confidence:** 3

**Summary:**

This paper proposes using large language models (LLMs) as policy makers. The authors argue that LLMs can address the challenge of processing large amounts of data quickly. They use a Stackelberg-Markov game to model the problem, where the principal plays against other players it influences. The authors use LLMs to model the principal. The core argument of the paper is that LLMs can offer a more sample-efficient solution to this problem.

**Strengths:**

1. The use of LLMs as economic policy generators is a novel and potentially more efficient approach compared to training agents from scratch.
2. The authors demonstrate their method can converge to effective policies with significantly fewer training iterations compared to baseline methods.
3. The paper leverages a well-defined formal framework (Stackelberg-Markov games) to analyze and compare the various policymaking methods.

**Weaknesses:**

1. The authors do not do any training or optimization process for the principal LLM. The LLM is indeed used to generate a policy, but there is no feedback loop to improve its choices based on the observed outcomes. The feedback is only used to change the game and the new observations this policy would see. The underlying assumption that an LLM can do this effectively is problematic, as LLMs are not trained for this, and therefore require additional fine-tuning or sampling methods. One example would be to use a Best-of-N (BoN) strategy, where several policy options are generated and the best one is selected based on some verifier. Here the authors would not be required to fine-tune an LLM, but they can still build a selector policy to choose from a discrete action space of candidates, and thereby improve the policy maker agent. This is the weakest point of the paper.
2. The paper doesn't explore the possible biases of the LLM in the context of the chosen environments. A robust evaluation would require testing how the LLM's biases affect the policies generated and, potentially, creating more diverse and sophisticated environments to test generalizability.
3. Overall presentation could be greatly improved. The authors describe related work and other algorithms throughout, and there is an underlying assumption that these methods are known to the reader. Many of the practical implementation details of the LLM-based approach are unclear. There is a lack of clarity regarding prompt engineering, parameter selection, and the specific LLM prompts used makes it difficult to evaluate the proposed method.  \
The Algorithm 1 block shows the general methodology that is used in previous work. Are the authors using that same methodology but only replacing the policy maker with an LLM? If that is the case, the authors should put more focus and emphasis on the LLM in the paper. There should be much more detail about how the LLM is being prompted, and potentially improved (see my first point above).

**Questions:**

See above

---

> ### Author Response · Authors · 2024-11-24
> **Rebuttal**
>
> *Optimization process...*
>
> We would like to kindly clarify a misunderstanding here. As outlined in Section 4, our method **does** have a feedback loop — we iteratively append observed outcomes to the prompt. This is optimization via in-context learning (ICL)[1,2] - allowing the LLM to approximate $\phi \mapsto u_o(\phi, \pi)$ without additional fine-tuning. Our results demonstrate that LLMs are able to utilize this feedback loop to optimize for these functions; however, we will revise our paper to include a more detailed explanation of the role of ICL within our method.
>
> *Possible biases...*
>
>  Can you expand on your meaning of bias in this context? If you mean bias in the context of fairness, then we agree; the biases of LLMs — politically, personally, or otherwise — is a matter of huge importance, as we express in Section 7.2. While we agree completely that rigorous testing of inherent biases must take place before the eventual implementation of any method in this field, we believe that such testing is outside of the scope of this paper and should be conducted alongside a diverse team constituting multiple perspectives and occupations outside of ML researchers.
>
> *Overall presentation...*
>
> We appreciate your opinion, but respectfully disagree. We understand that readers may not recognize methods such as AI Economist and MetaGrad, and have provided further information in Appendices A.1 and A.2, respectively. In terms of practical implementation of our method, we provide a detailed description in Section 4, as well as Figure 4, a comprehensive diagram of our method. Exact details regarding hyperparameter choice and selection process can be found in Appendices E.2 and E.3, and the 12 prompts used throughout our experiments can be found in Appendix H. As stated in Section 4, we do utilize the general methodology outlined in Algorithm 1 with an extension to incorporate historical data. Further detail on this extension is provided in Appendix D. To your point about placing more focus on the LLM within the paper, we believe that the main purpose of this work is to demonstrate the advantage that LLMs have over SOTA methods in this field. Therefore, we have chosen to emphasize our experimental results, which highlight the potential of LLM-based methods.
>
> [1] Qingxiu Dong, Lei Li, Damai Dai, Ce Zheng, Jingyuan Ma, Rui Li, Heming Xia, Jingjing Xu, Zhiyong Wu, Baobao Chang, Xu Sun, Lei Li, and Zhifang Sui. “A survey on in-context learning”, 2024. URL: https://arxiv.org/abs/2301.00234.
>
> [2] Shivam Garg and Dimiyris Tsipras and Percy S. Liang and Gregory Valiant. “What Can Transformers Learn In-Context? A Case Study of Simple Function Classes”, 2022, URL: https://papers.nips.cc/paper_files/paper/2022/file/c529dba08a146ea8d6cf715ae8930cbe-Paper-Conference.pdf
>
> We would like to **thank you** again for reviewing our work, and hope that our clarifications and responses have helped to address your concerns and critiques. We kindly request a reevaluation of our rating; we hope that our rebuttal to Weakness 1, which was qualified as the weakest point in the paper, may encourage you to raise your score. Thank you for your time.

---

### Official Review · Reviewer_YskH · 2024-11-04

**Soundness:** 3
**Presentation:** 2
**Contribution:** 2
**Rating:** 6
**Confidence:** 2

**Summary:**

This paper asks pre-trained LLMs to suggest a policy - for example to put incentives for cleaning up the river in the clean up environment - in such a way that individually trained RL agents trained on those tasks achieve high social welfare. The LLM can also observe the feedback of its policy and modify it in an iterative manner. For example, if the incentive to clean the river and abstain from harvesting apples is not enough, the LLM may increase the incentive in the next rounds. The authors find that while the LLM has a bias to do changes slightly and therefore slightly underperforms previous RL based methods, it can do so by observing much fewer samples from the environment, therefore being more sample efficient.

**Strengths:**

It is always a good idea to try the simple solutions on a benchmark and show how they perform. I think this is the main strength of the paper. What if instead of the complex RL mechanisms, we just ask a pre-trained instruct-tuned LLM to solve the task. If this outperforms all other methods, it is a valuable contribution to that field. I think that is the main strength of the paper.

In detail, this paper has three messages : 1) If we remove the conditioning of the policy on the state, other methods performance remains the same, suggesting the complexity was not needed at the first place 2) Asking an LLM to suggest the policy, and then refine the policy, works quite well and also does need much fewer samples 3) Asking the LLM to do so, has its drawbacks, for example, the LLM tries to suggest a conservative incentive even when a stronger one is needed.

The paper is original in the sense that I have not seen a paper using LLMs to suggest policies before. It is also clearly written. However, the significance of the main message being the sample efficiency of the LLMs I think is a bit low which I will elaborate on in the weakness section.

I think the main important consequence of the paper in the field is that we need much better benchmarks when discussing solving social dilemmas by LLMs. The current environments are toy-ish because we don't have the correct RL tools to solve even them. However, this paper shows that if we want to explore social dilemmas using LLMs as policy-makers, which is a valid approach, we don't have the benchmarks which their difficulty actually matches the strengths of these neural networks. We need environments which are large, hard to describe in short sentences, and need much more iterations to get the policy right.

**Weaknesses:**

The paper argues that using pre-trained LLMs is sample efficient. I have two main concerns about this statement.

1) LLMs are not that sample efficient:
The paper argues that using pre-trained instruction-tuned LLMs is more sample efficient. But, I think we should also take into account all the pre-training and data the feeds into these models. Taking them into account, I don’t think using LLMs is sample efficient.

2) LLMs have seen these environments possibly:
I am not sure of these. But for example, by asking chatgpt 4o at Nov 3rd of 2024 I can confirm that by just asking “do you know the RL clean up environment and what a good strategy would look like without search?” it can reiterate what the clean up environment is. Therefore, it is possible that they already know what the dilemma of the environment is and how to solve it in the sense that it is discussed in their pre-training data. I don’t know if this is what is actually wanted, because it is okay for example to look at how other countries fixed their economies when trying to propose a policy to another country. But also, these environments are so toyish that knwoing what the dilemma is, is most of the answer. Therefore, it is not clear if doing the same to complex environments or unseen environments shows the same sample complexity.

The other important problem is, we are using the LLM as the learning algorithm here in the sense that we will put all the environment details and actions and previous proposed policies in the context. I am wondering if this is a scalable approach? Especially, when environments get larger, and the require optimization iterations steps also grow a lot, I think two issues arise. First, the prompt gets very long. Second, we need much more iterations to optimize the proposed policy.

Also, at the end, the best final policies are reached by RL methods and not the LLMs, suggesting that the LLMs are not very good at imitating a learning algorithm, although their first guesses are much more educated which I think given their massive pre-training data and possible exposure to these environments, makes a lot of sense.

I proposed a marginally behind acceptance threshold because I think the focus of the writing of the paper, the sample efficiency of this approach, is not significant. I think if the paper was written as an analysis of what happens if we try to use LLMs to solve current benchmarks social dilemmas, its contribution would be much more clear. The paper would discuss the attempts, the limitations, etc.

**Questions:**

1- Should we take the huge data, instructions, compute and RL post-training that these models needed into account when assessing if they are sample efficient?

2- There is a possibility that some texts discussing the environments may be on internet suggesting what is an optimal approach especially in the papers written in the field. Is there any way to assess this?

3- What would be the scalability properties of the paper's paradigm? Is there a chance that it scales well to more complex settings?

---

> ### Author Response · Authors · 2024-11-24
>
> *Sample efficient..., Question 1*
>
> We appreciate this perspective. While LLMs do require substantial data and computational resources to pretrain, our focus is on sample efficiency when adapting to these problem settings, which is the number of samples to converge on a new environment. The LLMs utilize in-context learning (ICL) to continue refining their responses past their initial educated guess. Importantly, these LLMs were not fine-tuned for simulated policymaking — we made no adaptations and conducted no additional training from environment to environment. This generalizability speaks to the potential of LLMs to fulfill a broader, more adaptable policymaking role, in stark contrast to RL-based methods without such a clear path to generalizability. Lastly, the broad continued development of LLMs points to the strong possibility that these pre-trained models will continue to be accessible.
>
> *LLMs seeing environment...*
>
> We would like to clarify that, as shown in Appendix H, the prompts provided to the LLM **do not reference the name** of either preexisting environment (those being Commons Harvest Open and Clean Up) or any specific details that would allow the models to directly identify them. Additionally, these environments were altered to enable the addition of principals, which cannot have been in the training set as we are the first to make these alterations.
>
> Furthermore, we would argue that the elements of selfishness and ‘short-terminism’ found within our experiments are relevant and reflect common dilemmas seen in the real world; in fact, Commons Harvest Open is named for the prominent economic theory of the ‘tragedy of the commons’. As such, we believe that frontier LLM models would be exposed to these dilemmas regardless of the Clean Up and Commons Harvest Open environments presence within their training data. In regards to your final point, we acknowledge that grasping the social dilemma certainly plays a role in how the LLM is able to initially respond, as evidenced by their zero-shot performances in Figure 4. However, past this initial iteration, the LLMs show a clear ability to continue refining their outputs via the usage of ICL. We point out that if the success of the LLM was entirely due to their pre-training, they would show no improvement; while the initial outputs are strong in comparison to baselines, they are significantly weaker than the converged outputs of most included methods, with approximate payoffs of 25, 21, and 15 on Harvest, Clean Up, and CER, respectively.
>
> *Scalability..., Question 3*
>
> Thank you for your question. In fact, we believe that the LLM-based approach would scale better than existing RL-based methods to more complex settings due to our sample efficiency gains. To your point about the length of the prompt, we do not foresee this as an issue. The context window and recall power of frontier LLMs is remarkable, as exemplified by this experiment [1] by Anthropic testing Claude 2.1s 200k token context window.  As for the iterations necessary to optimize a more complex policy, we ask you to consider the fact that as LLMs continue improving, the cost of inference will continue to fall [2], especially when paired with advancing hardware [3] and software [4,5]. We believe that the main computational bottleneck blocking tractability will be complex multi-agent simulation; this bottleneck is not specific to our method, and we predict that engineering advancements will help to mitigate this problem.
>
> *Final policies...*
>
> We believe your critique of our final performance may be overstated. We’d like to politely refer you to Table 1, showcasing our main results across our three environments. The primary RL-based methods are AI Economist and MetaGrad, as opposed to the bandit based methods of UCB, Thompson Sampling, and epsilon greedy. Note that the bandit methods are not scalable to more complex problem settings, as the action spaces become combinatorially large. We outperform MetaGrad on each environment, and we provide a detailed analysis of this in Appendix A.2. In addition, we would like to call your attention to Harvest, where GPT-4o mini outperforms AI Economist, and CER, where both LLMs outperform AI Economist. While AI Economist outperforms both LLMs on Clean Up, we believe that overall these constitute very strong final results, especially when considering that  the LLMs were dozens of times more sample efficient.
>
> As previously addressed, we politely disagree as to your point about LLMs only imitating learning algorithms, and point you towards the literature on ICL[6,7], which show that LLMs can learn without updating their parameters. However, we will revise our paper to include a deeper discussion of the role of ICL within our method.

---

> ### Author Response · Authors · 2024-11-24
>
> *Marginally behind acceptance...*
>
> Thank you for explaining the rationale behind your score. While we believe that our sample efficiency gains are significant as detailed above, we do discuss the motivations and advantages of using LLMs to solve social dilemmas throughout the paper. In Section 3, we use Algorithm 1 to analyze and highlight misunderstandings in current methods as motivation for the introduction of LLMs. Furthermore, we perform ablations on unique advantages offered by LLMs; these ablations, focused on contextualization and historical observations, can be found in Section 5.3 and Appendix B, respectively. As to your point about discussing the limitations, we kindly refer you to Section 7.1.
>
> *Question 2*
>
> Yes, there exist methods to identify if certain text was present in the training data for an LLM, as this is important for model security and the ethics of privacy regarding training data. Examples include [8] and [9].
>
> [1] Anthropic, “Long Context Prompting for CLaude 2.1”, 2023, URL: https://www.anthropic.com/news/claude-2-1-prompting
>
> [2] BoredGeekSociety, “OpenAI Model Pricing Drops by 95%”, 2024, Medium, URL: https://medium.com/@boredgeeksociety/openai-model-pricing-drops-by-95-3a31ab0e04e6#:
>
> [3] Dennis Abts et. al, “A Software-defined Tensor Streaming Multiprocessor for
> Large-scale Machine Learning”, 2022, URL: https://groq.com/wp-content/uploads/2023/05/GroqISCAPaper2022_ASoftwareDefinedTensorStreamingMultiprocessorForLargeScaleMachineLearning-1.pdf
>
> [4] Iz Beltagy and Matthew E. Peters and Arman Cohan, “Longformer: The Long-Document Transformer”, 2022, URL: https://arxiv.org/pdf/2004.05150
>
> [5] Tri Dao and Dan Fu and Stefano Ermon and Atri Rudra and Christopher Ré, “FlashAttention: Fast and Memory-Efficient Exact Attention with IO-Awareness”, 2022, URL: https://proceedings.neurips.cc/paper_files/paper/2022/file/67d57c32e20fd0a7a302cb81d36e40d5-Paper-Conference.pdf
>
> [6] Qingxiu Dong, Lei Li, Damai Dai, Ce Zheng, Jingyuan Ma, Rui Li, Heming Xia, Jingjing Xu, Zhiyong Wu, Baobao Chang, Xu Sun, Lei Li, and Zhifang Sui. “A survey on in-context learning”, 2024. URL: https://arxiv.org/abs/2301.00234.
>
> [7] Shivam Garg and Dimiyris Tsipras and Percy S. Liang and Gregory Valiant. “What Can Transformers Learn In-Context? A Case Study of Simple Function Classes”, 2022, URL: https://papers.nips.cc/paper_files/paper/2022/file/c529dba08a146ea8d6cf715ae8930cbe-Paper-Conference.pdf
>
> [8] Pratyush Maini and Hengrui Jia and Nicolas Papernot and Adam Dziedzic. “LLM Dataset Inference: Did you train on my dataset?” 2024. URL: https://arxiv.org/abs/2406.06443
>
> [9] Justus Mattern and Fatemehsadat Mireshghallah and Zhijing Jin and Bernhard Schölkopf and Mrinmaya Sachan and Taylor Berg-Kirkpatrick. “Membership Inference Attacks against Language Models via Neighbourhood Comparison” 2024. URL: https://arxiv.org/abs/2305.18462
>
> **Thank you** for your comprehensive review. We are pleased to hear that you found our paper original and well written, and that you see our work as a valuable contribution to the field. We are gratified by the time and effort that was clearly put into our work, and believe that your feedback will help us to improve our research. We feel confident that our work will make an impact on this field and drive future researchers to continue advancing LLM-based methods. We encourage you to revisit your evaluation of our work, and we look forward to any further questions you may have.

---

> ### Comment · Reviewer_YskH · 2024-12-01
> **Response to Rebuttal**
>
> Dear Authors,
>
> Thank you for your response to my rebuttal. I understand your viewpoint that it is a very important baseline to see what an LLM could do in these cases as a policy-maker. I increase my score from 5->6 just to reflect that I think it is always important to compare complicated RL algorithms to just using a pre-trained LLMs. However, I cannot go upper than that because at the end the RL methods seem to be superior and I think the sample efficiency of the pre-trained LLMs is overstated in your paper as they are trained on huge amount of data.

---

### Official Review · Reviewer_ULcW · 2024-11-05

**Soundness:** 3
**Presentation:** 3
**Contribution:** 2
**Rating:** 5
**Confidence:** 3

**Summary:**

This paper presents a method for economic policymaking that utilizes LLMs to overcome the sample inefficiencies found in current RL models. By incorporating LLMs into MALR environments, the authors showcase improvements in efficiency compared to existing RL-based models, such as AI Economist. The approach integrates contextual and historical data to optimize output in three test environments, achieving faster convergence and comparable or better performance than other methods.

**Strengths:**

1. The paper clearly outlines its motivation, highlighting that current SOTA methods face sample inefficiencies. This has led to the exploration of using LLM agents for policymaking.

2. The paper is well-structured, with a clear articulation of its methods and contributions.

3. Extensive experiments across multiple environments effectively validate the proposed approach with thorough and convincing results.

**Weaknesses:**

1. While using a Stackelberg game model is reasonable and straightforward for this policy-making problem, the observation that RL-based methods are unaffected by state observations raises questions about the environment’s simplicity; this suggests the need either for testing in environments where observations impact performance or for confirming that this lack of impact holds consistently across all environments.

2. The paper does not clarify whether LLM success arises from pre-trained strategies or from actual understanding of the policy-making environment.

3. LLMs may benefit from pre-training on diverse datasets, which could give them an edge over RL methods that train from scratch, raising questions about the fairness of comparisons.

4. While LLMs show rapid convergence, they often do not match the higher final performance achieved by RL-based methods in certain environments.

**Questions:**

1. One of this paper's major findings is that state observations don't affect performance in current SOTA methods, which is surprising and interesting; however, if observations are irrelevant to performance, why model the problem as a Stackelberg game which inherently assumes strategic interaction based on observation? This suggests the current policy-making might be reducible to a simpler fitting problem rather than requiring a complex game-theoretic framework, especially in relatively simple environments.

2. The paper's approach uses a simple LLM implementation yet achieves impressive results with rapid convergence. Whether LLMs have already acquired optimal policy-making strategies during pre-training or if they understand Stackelberg equilibrium concepts from the prompt.

3. As the LLM approaches benefit from pre-training on diverse data, while the RL methods start from scratch (especially for the Meta-Griednat method, which needs huge exploration during the training). It is unclear whether LLM’s performance truly represents a better decision-making approach or reflects its pre-training advantage.

4. The experimental results show that while RL-based methods require significantly more time to converge, they ultimately achieve better final performance than LLM-based approaches in several environments (particularly noticeable in the clean-up environment). Does this suggest that RL-based methods, despite their slower convergence, have a higher performance ceiling than non-finetuned LLM approaches? Or, the LLM-based method may be limited in its ability to find truly optimal policies without fine-tuning.

---

> ### Author Response · Authors · 2024-11-24
> **Rebuttal**
>
> *State observations...*
>
> We would like to clarify a possible misunderstanding here — the environments used in the ablations in Section 2 are not the same as those used in our main results. It is likely the case that our environments would in fact require state observations. In particular, we would like to kindly highlight our third environment described in Section 5.1. State observations play a crucial role in Contextual Escape Room, as without knowledge of which lever is activated, a principal only would be able to provide the optimal incentive a third of the time on average.
>
> *LLM success..., Question 2*
>
> Both factors contribute to the success of the LLM. Figure 4 demonstrates that the LLMs are able to leverage their pre-training to contextualize their very first output, such that their zero-shot capabilities are unsurprisingly greater than RL-based methods that need to be retrained for each environment. However, past the initial iteration, the LLMs show a clear ability to continue refining their outputs via the usage of in-context learning (ICL)[1,2]. We point out that if the success of the LLM was entirely due to their pre-training, they would show no improvement past the first generation. We will revise our paper to more clearly reflect this, and provide a more detailed analysis of the ICL process.
>
> *Fairness of comparisons..., Final performance..., Question 3, 4*
>
> Thank you for raising these points.
>
> Regarding the advantage from pre-training: We in fact count the pre-training as an advantage to our method, allowing the principal to generalize and take in a wide range of contextualizing text-based inputs. Additionally, the RL-based methods are allowed to update their weights, whereas our LLM-based method’s weights are frozen, and is restricted to using just ICL. This can also be seen as an unfair advantage for the RL-based methods. While the pre-training of the LLM may not be entirely fair, we believe that this advantage is, and will continue to be, a practical reality; pre-trained LLMs are accessible, applicable, and continuously improving.
>
> Regarding fairness of comparison: While the RL methods start from scratch and lack generalizability, they are specifically tuned for each environment and should be expected to perform well after extensive training. Arguably, as the leading method in this field, AI Economist should be expected to outperform LLMs with no fine-tuning after training for dozens of times longer. However, this does not seem to be the case, as detailed below.
>
> Regarding final performance: We believe your critique of our final performance may be overstated. We’d like to politely refer you to Table 1, showcasing our main results across our three environments. The primary RL-based methods are AI Economist and MetaGrad, as opposed to the bandit based methods of UCB, Thompson Sampling, and epsilon greedy. Note that the bandit methods are not scalable to more complex problem settings, as the action spaces become combinatorially large. We outperform MetaGrad on each environment, and we provide a detailed analysis of this in Appendix A.2. In addition, we would like to call your attention to Harvest, where GPT-4o mini outperforms AI Economist, and CER, where both LLMs outperform AI Economist. While AI Economist outperforms both LLMs on Clean Up, we believe that overall these constitute very strong final results, especially when considering that the LLMs were dozens of times more sample efficient.
>
> *Why model as a Stackelberg game...*
>
> Thank you for your question. Notably, we do not believe that the ineffectuality of SOTA state observations means that state observations are always unimportant; rather, that the existing methods had taken a misguided approach to incorporating them at this level of environment complexity. We model the problem as a Stackelberg game to leave open the possibility of the aforementioned strategic interaction based on observation for scalability to observation-dependent environments. In CER, for example, state observations are crucial. We kindly refer you to Appendix B, where we further explore their relevance and compare the effectiveness of our method to MetaGrad and AI Economist in leveraging the information.
>
> 2. [1] Qingxiu Dong, Lei Li, Damai Dai, Ce Zheng, Jingyuan Ma, Rui Li, Heming Xia, Jingjing Xu, Zhiyong Wu, Baobao Chang, Xu Sun, Lei Li, and Zhifang Sui. A survey on in-context learning, 2024. URL https://arxiv.org/abs/2301.00234.
>
> 3. [2] Shivam Garg and Dimiyris Tsipras and Percy S. Liang and Gregory Valiant. “What Can Transformers Learn In-Context? A Case Study of Simple Function Classes”, 2022, URL: https://papers.nips.cc/paper_files/paper/2022/file/c529dba08a146ea8d6cf715ae8930cbe-Paper-Conference.pdf

---

> > ### Comment · Reviewer_ULcW · 2024-11-24
> >
> > Thank you for your detailed response. However, after reviewing your feedback to all reviewers, I still have some concerns:
> >
> > 1. Regarding the performance results - I maintain my position and note that other reviewers (e.g., **Reviewer YskH**) share similar concerns. While you achieve comparable results on CER and Harvest, the claim of sample efficiency needs to be carefully examined. As **Reviewer YskH** pointed out, if we're discussing data efficiency, we should account for the massive pre-training data and compute resources required by LLMs. I suggest expanding your evaluation to include more diverse and complex environments to better demonstrate the method's capabilities and generalization.
> >
> >
> > 2. On your claim about the method benifit from the ICL: While I note that you adopted a CoT-like format for organizing historical observations (line 254-269), the paper employs relatively basic ICL techniques. This raises several questions:
> >
> >     2.1 ICL is an inherent capability of LLMs rather than a novel contribution of your method.
> >
> >     2.2 Why did you choose this specific format over other ICL approaches? Were others considered or tested?
> >
> >     2.3 As **Reviewer SeUt** suggested, why not consider fine-tuning approaches that allow direct interaction with the environment?
> >
> > 3. The paper still doesn't conclusively demonstrate whether the LLM's success comes from pre-trained knowledge or actual understanding of policy-making dynamics. Direct ablation studies would help clarify this.

---

> > > ### Author Response · Authors · 2024-11-27
> > >
> > > We thank you for taking the time to go over our response and voice your concerns.
> > > 1. While it is certainly true that LLMs require a massive amount of pretraining, this phase is a one-time cost that has already been paid. We ask you to consider the comparison of assembly language and modern compilers; though these compilers have taken substantial resources to create, their existence now means that coding is much faster. As with LLMs, we would argue that the initial resource cost does not invalidate the resultant efficiency gains.
> > > 2. We would like to clarify that we do not see or claim ICL as a contribution of our method. Though it may seem tempting to add as much complexity as possible to a method for it to appear to contribute more, our decision to use the simplest possible ICL strategy with no fine-tuning was deliberate: that the seminal works of such an impactful field be outperformed on several occasions by this should be highlighted as starkly as possible. Why would we add expensive fine-tuning that further increases the pretaining sample count you have raised concerns about, and decreases the reproducibility of our results?
> > > 3. Thank you for making this point. The answer is both - pretraining allows the model to contextualize its first response, but we see clear increases past this initial iteration. This is not surprising, as in-context learning is a well-documented phenomenon. Of course, the LLM does not have any “actual understanding of policy-making dynamics” - it is merely a probabilistic model of language.

---

> ### Author Response · Authors · 2024-11-24
>
> **We thank you** for your consideration of our work, and are pleased to hear that you found our paper well-structured and that we effectively validated the proposed approach. We hope that we have sufficiently addressed all of your points and await any further questions you may have. We hope that with our clarifications, you may consider reevaluating our work.

---

### Official Review · Reviewer_tzs7 · 2024-11-10

**Soundness:** 3
**Presentation:** 2
**Contribution:** 2
**Rating:** 5
**Confidence:** 3

**Summary:**

The paper proposes a prompting-based agent framework for Economic policy-making problems. The paper first illustrates the over-complicated problems with previous RL-based methods. To address this issue, the authors leverage few-shot prompting and derive an LLM-based policy-making agent. Empirical evaluations demonstrate that their LLM-based method outperforms traditional RL-based baselines across various multi-agent environments.

**Strengths:**

Strengths:
1. The overall writing is clear to follow.
2. The paper offers a comprehensive overview of the background, previous methods, and proposed methods.
3. The paper validates the performance in 3 environments and the ablation study seems to be sufficient.

**Weaknesses:**

1.Prompting-based agents and techniques have been applied in a lot of fields (beyond economics) and the proposed method seems to be not that novel. Techniques like memory, reflection, are proposed and used in many other agent frameworks.

2. A lot of related work and citations are missing. For instance, the proposed algorithm seems to refer to ReACT [1], Reflextion [2], AutoGPT [3] and more agent frameworks.

3. The writing logic is a bit messy, especially for section 3. I understand that section 3 is for analyzing the overcomplicating issue of previous methods. But it seems not that related to section 4 and cannot help strengthen your major proposed algorithm. I’d suggest try to move the main discussion into the appendix to improve writing fluency.

4. I am not an expert in this domain, but from the prompt example shown in the appendix, the problem the principle agent tries to solve has nothing to do with the multi-agent setting. It is, in fact, a bandit optimization (so just one-step decision-making by keeping all other follower agents as part of the environment). Also, these three environment settings look very similar. These two issues largely weaken their effectiveness in showing the algorithm’s performance on complex and general problems.

[1] Yao, Shunyu, Jeffrey Zhao, Dian Yu, Nan Du, Izhak Shafran, Karthik Narasimhan, and Yuan Cao. "React: Synergizing reasoning and acting in language models." arXiv preprint arXiv:2210.03629 (2022).
[2] Shinn, Noah, Federico Cassano, Ashwin Gopinath, Karthik Narasimhan, and Shunyu Yao. "Reflexion: Language agents with verbal reinforcement learning." Advances in Neural Information Processing Systems 36 (2024).
[3] Yang, Hui, Sifu Yue, and Yunzhong He. "Auto-gpt for online decision making: Benchmarks and additional opinions." arXiv preprint arXiv:2306.02224 (2023).

**Questions:**

1.Is there any other LLM-based baseline you can include to strengthen your comparison?
2.The MetaGrad baseline seems weird? Why it cannot improve in all environments.
3.The experimental results seem to be repetitive? Table 1 and Figure 1 seem to be the same results?

---

> ### Author Response · Authors · 2024-11-24
> **Rebuttal**
>
> *Prompting-based agents...*
>
> Thank you for your observation. However, the application of LLMs to the problem setting of MARL environment optimization is non-trivial and required careful design of the state-action spaces and reward functions. Due to our careful observation of the over-complication of the problem setting by the previous methods such as AI Economist and MetaGrad, we are able to propose a new theoretical model (Algorithm 1) for unifying previous work, which allows us to simplify previous representations of state-action spaces and enables the usage of LLMs. In our experiments, we make the design choices of the inclusion of natural language context for processing auxiliary observations, the removal of agent non-stationarity, and the image input used by the previous methods. None of these choices are obvious, and we show in our ablations that the removal of the context is significantly detrimental to our method’s performance. In addition, we believe that the order of magnitude improvement in sample efficiency in comparison to SOTA methods reveals the potential of LLMs to succeed in this field broadly, and we are confident that our work will serve as inspiration for future LLM-based policymakers.
>
> *Related work...*
>
> We appreciate you bringing this to our attention, and apologize for our oversight. We have carefully looked through these papers and added the relevant citations in Section 3.
>
> *Writing logic...*
>
> Thank you for your feedback. We will take it into consideration in our revisions of the paper.
>
> *Bandit optimization ..., environments...*
>
> You are correct that the environments are formalized as a bandit optimization problem. However, this was by design in an attempt to represent the problem setting simply and cohesively as explained above. In this effort, we share motivation with works such as [1], whose findings suggested that simple and effective  algorithms are often overlooked. We draw from Gerstgrasser & Parkes [2] to do so, framing the bilevel optimization problem as a Stackelberg-Markov game such that the learning problems of the principal and follower can be separated. This framing allows us to shed light on the overcomplications within existing methods as described in Section 3, without losing applicability to the problem setting.  We respectfully disagree that this weakens our effectiveness in showing the algorithm’s performance — in fact, this enables us to solve more complex problems by disentangling the agents’ and principal’s learning problems. As to your point about the similarity of the environments, we would appreciate more specification; we believe that our environments, pictured in Appendix G, encompass a range of social dilemmas, and principal actions are unique across each environment.
>
> *Other LLM-based baseline...*
>
> Thank you for your question. Could you clarify what you mean by ‘LLM-based baseline’? If you are inquiring about a different method entirely, then the answer is no. To our knowledge, ours is the only LLM-based method in this relatively young field for now; as we address above, we hope this work inspires other researchers to further develop LLMs towards policymaking.
>
> *MetaGrad baseline...*
>
> Thank you for asking for clarification. However, we would like to clarify that we provide a detailed analysis of the poor performance of MetaGrad in Section 5 and Appendix A.3.
>
> *Repetitive experimental results...*
>
> We believe that you are referencing Table 1 and Figure 4, and you are correct — Table 1 is introduced as a numerical representation of the results pictured in Figure 4, and was included for quantitative baselines as a comparison for future work. Without Table 1, future researchers would only be able to observe a rough estimate of the results.
>
> [1] Matthias Gerstgrasser and David C. Parkes, “Oracles & Followers: Stackelberg Equilibria in Deep Multi-Agent Reinforcement Learning”, 2023, URL: “https://proceedings.mlr.press/v202/gerstgrasser23a.html”
>
> [2] Scott Fujimoto and Shixiang Shange Gu, “A Minimalist Approach to Offline Reinforcement Learning”, 2021, URL: https://proceedings.neurips.cc/paper_files/paper/2021/file/a8166da05c5a094f7dc03724b41886e5-Paper.pdf
>
> **Thank you** for your detailed review of our work. Your feedback has proven helpful, and we look forward to revising our paper to strengthen our work. Given these clarifications, we hope that you may consider raising your score. We look forward to any further questions or feedback.

---

### Author Response · Authors · 2024-11-24
**Global Note**

We are grateful for all of the thoughtful feedback and comments provided by the reviewers. We were pleased to hear that reviewers found the paper “well-structured, with a clear articulation of its methods and contributions” (ULcW), and that we leveraged a “well-defined formal framework to analyze and compare the various policymaking methods” (SeUt). Further, we are glad to hear that our “extensive experiments across multiple environments effectively validate the proposed approach with thorough and convincing results”(ULcW), and agree with the assessment of Reviewer YskH who stated that “if this outperforms all other methods, it is a valuable contribution to that field”.

---

### Meta-Review · Area_Chair_4bAR · 2024-12-20

**Metareview:**

This paper presents an agent framework, leveraging prompting techniques for multi-agent systems. While the writing is clear and the paper provides a thorough overview of background and methods, with some empirical validation, it ultimately falls short of conference standards due to a lack of novelty and questionable experimental design.  The core weakness lies in the incremental nature of the proposed approach. The techniques employed, such as memory and reflection, are already established within the agent literature, and the paper fails to demonstrate significant advancement beyond existing work like ReAct, Reflexion, and AutoGPT, which are surprisingly not cited. Additionally, the paper suffers from organizational issues, with Section 3 feeling disjointed and detracting from the main narrative.  Most concerning, however, is the validity of the experimental evaluation. The chosen environments appear overly simplistic and redundant, essentially reducing the multi-agent problem to a single-agent bandit optimization task, casting doubt on the generalizability and effectiveness of the proposed framework in complex settings

**Additional Comments On Reviewer Discussion:**

Reviewer's responses have been partially addressed during rebuttal, however the paper will still be benefitted with an additional iteration.

---

### Decision · Program_Chairs · 2025-01-22

Reject